# Subaerial and subglacial seismic characteristics of the largest measured jökulhlaup from the Eastern Skaftá cauldron, Iceland

Eva P. S. Eibl[1], Kristin S. Vogfjörd[2], Benedikt G. Ófeigsson[2], Matthew J. Roberts[2], Christopher J. Bean[3], Morgan T. Jones[4], Bergur H. Bergsson[2], Sebastian Heimann[1], and Thoralf Dietrich[1]

[1]University of Potsdam, Institute of Geosciences, Karl-Liebknecht-Str. 24-25, 14476 Potsdam, Germany
[2]Icelandic Meteorological Office, Bústaðavegi 7–9, 108 Reykjavík, Iceland
[3]Geophysics Section, School of Cosmic Physics, Dublin Institute for Advanced Studies, 5 Merrion Square, Dublin 2, Ireland
[4]Centre for Planetary Habitability (PHAB), Department of Geosciences, University of Oslo, PO Box 1028, Blindern 0315 Oslo, Norway

**Correspondence:** Eva Eibl (eva.eibl@uni-potsdam.de)

**Abstract.** Subglacial floods cause seismic tremor that can be located and tracked in space and time using a seismic array. Here, we shed light on the generating mechanisms of the seismic signals observed during the largest measured flood from the Eastern Skaftá cauldron in the Vatnajökull ice cap, Iceland. We track the propagation of the flood in 2015 using two seismic arrays and a local seismic network in combination with GPS, hydrological and geochemical data. We find that as the water drained from the subglacial lake beneath the cauldron, families of icequakes were generated in the area around the cauldron while the glacier surface gradually subsided by more than 100 m. We detected several-hours-long, non-harmonic tremor and high-frequency transient events migrating downglacier following the subglacial flood front. We suggest that this tremor is composed of repeating, closely spaced icequakes generated as the glacier was being lifted, cracked, and deformed enabling the subglacial water flow. When the lake had largely drained, the pressure within the underlying hydrothermal system dropped. At this time, we recorded minute-long tremor bursts and hour-long harmonic tremor emanating from the cauldron area. We interpret these as caused by hydrothermal explosions in the geothermal system within the cauldron and as vigorous boiling in the crustal rocks, respectively, an interpretation corroborated by floodwater geochemical signals. Finally, the flood also led to detectable tremor due to more energetic flow in the rapids near Sveinstindur in the Skaftá river. We conclude that the flood generated five different seismic signal types that can be associated with five different geophysical processes, including the wide spectrum from brittle failure to explosions, boiling and turbulent flow.

## 1 Introduction

Subglacial volcanic and geothermal systems beneath glaciers cause a substantial flood hazard in areas surrounding glaciers (Waythomas et al., 2013; Björnsson, 2003; Magnússon et al., 2012; Roberts, 2005; Cook et al., 2018; Eibl et al., 2020). Glacial outburst floods, termed jökulhlaups in Icelandic, can occur through steady melting above ice-covered geothermal areas (Björnsson, 2003, 2010), through melting by magma–ice interaction during a volcanic eruption (Björnsson, 2003; Gudmundsson et al., 1997; Sturkell et al., 2008), as well as through the release of water stored in subglacial or marginal lakes dammed by glaciers

(Björnsson, 1976; Roberts et al., 2005; Bartholomaus et al., 2015; Grinsted et al., 2017; Lindner et al., 2020; Livingstone et al., 2019; Behm et al., 2020) or moraines (Cook et al., 2018). Improving the understanding of source processes and generation mechanisms of subglacial floods is challenging as (i) the flood has to be detected beneath several hundred meters of ice; (ii)

instruments are either difficult to maintain on the ice and nunataks within the ice or they are located outside the glacier, far from the signal generating source; and (iii) seismic signals accompanying a flood are often weak, non-impulsive, long-lasting, lack discernible seismic phases and are therefore intrinsically difficult to analyse to extract source location and mechanism.

In close proximity to rivers it has been found that long-lasting seismic signals referred to as tremor, are generated both by turbulent flow and bedload transport in the rivers (Burtin et al., 2011; Gimbert et al., 2014, 2016; Schmandt et al., 2013). The

tremor amplitude correlates with the discharge (Hsu et al., 2011; Burtin et al., 2008), as was recently confirmed in a glacial environment by Bartholomaus et al. (2015) and Gimbert et al. (2016), where Bartholomaus et al. (2015) correlated the tremor amplitude at 1.5 to 10 Hz measured at 1 to 5 km distance with the discharge, while Gimbert et al. (2016) explained the tremor amplitude between 2 to 12 Hz as being due to turbulent flow interacting with the bed roughness. However, they both suggest that at more than 1 km distance, the seismic signal caused by turbulent water flow dominates over the signal caused by bedload

transport.

At larger distances the seismic signal linked to turbulent water flow still needs to be distinguished from other possible tremor sources. In a glacial environment, there are a variety of possible tremor sources (Podolskiy and Walter, 2016), including resonating water filled cracks or channels (Röösli et al., 2014; Chapp et al., 2005; Winberry et al., 2009; Heeszel et al., 2014; Lindner et al., 2020), englacial water flow in a moulin (Röösli et al., 2014, 2016; Lindner et al., 2020), regularly repeating

icequakes (MacAyeal et al., 2008; Winberry et al., 2013; Müller et al., 2005; Lindner et al., 2020; Behm et al., 2020; Lipovsky and Dunham, 2016) and hydrothermal boiling (Leet, 1988; Montanaro et al., 2016). The tremor signals from these different sources need to be characterized and distinguished from flood-related tremor, observed during subglacial floods, as some might indicate hazardous migrating processes, whilst others might be stationary, non hazardous, 'normal background' processes. Furthermore, despite various reports on seismic signals such as quakes and tremor before and during subglacial floods (Winberry

et al., 2009; Bartholomaus et al., 2015; Lindner et al., 2020; Behm et al., 2020), a thorough and continuous tracking of flood migration beneath the ice from the draining lake to the glacier terminus was only recently achieved by Eibl et al. (2020) for 4 subglacial floods in Iceland. At 10 to 52 km distance they detect a migrating tremor source with peak frequency at 1.3 Hz.

Iceland is an ideal place to study subglacial flood-related seismic signals as there are multiple subglacial floods per year which produce detectable signals such as tremor (Eibl et al., 2020). In several glaciological, geomorphological or hydrological studies

(Böðvarsson et al., 1999; Einarsson, 2009; Einarsson et al., 2016; Old et al., 2005; Roberts et al., 2003), researchers used one-minute averages of filtered seismic signals to exhibit the evolution and characteristics of different jökulhlaups. These filtered signals suggest that the seismic character of subglacial floods changes with time, where in particular floods draining subglacial high-temperature geothermal areas exhibit the following characteristics: a period of quakes, intermixed with or followed by weak tremor that is followed by strong tremor bursts. If no network stations are located near the flood path, then the weak

tremor is usually not detected above the background noise. Due to a lack of continuous data, sparse seismic networks and uncertainties in the timing of the subglacial propagation of these floods, there has so far been little in-depth, seismological

analysis of this type of activity and the processes that might generate it. A comprehensive overview of the different sources generating seismic signals in glacier environments and various commonly applied seismic analysis methods can be found in Podolskiy and Walter (2016).

In 2013 we installed a multidisciplinary network at Vatnajökull glacier, consisting of seismic arrays outside the glacier margin, and seismic and GPS stations within the glacier. The network was specifically designed to (i) monitor subglacial floods from west Vatnajökull, and (ii) used to demonstrate a real-time early-warning capability. Eibl et al. (2020) tracked four subglacial floods with peak discharges in the range of 210 to 3000 $m^3/s$ using one seismic array, which is a cluster of closely spaced sensors capable of resolving direction to the tremor source and the apparent speed of the waves across the array. They con-

cluded that large floods travel faster than smaller floods and that seismic arrays can be used to detect the floods several hours in advance of the hydrological network located in the area outside the glacier margin, allowing an improved early-warning of on-coming floods. The study however did not delve into detailed examination of the different types of seismic events associated with the floods which would be crucial for a better understanding and robust interpretation of the signals.

Here, we focus on an in-depth analysis of the largest of the four subglacial floods, a flood emanating from the eastern Skaftá

cauldron in Vatnajökull in 28 September to 2 October 2015, with the goal of characterizing the different seismic signals associated with the flood to understand the generating mechanism, aided by analysis of other multidisciplinary data, such as geodetic observations of the glacier movements and its response to the subglacial flood, hydrological and geochemical data recording the volume, flow and dissolved chemicals in the flood water (section 3). We track the flood both beneath the glacier and in the affected glacial river. We analyse icequakes, 4 tremor sources and one noise source detected in seismic data from the local seis-

mic network, SIL, and the two arrays in the context of the geochemical data (section 4.1). We describe the located icequakes (section 4.2), weak persistent tremor (section 4.3), transient high-frequency events (section 4.4), strong tremor bursts and harmonic tremor tails (section 4.5) and the tremor and noise of the river (section 4.6) in detail. We discuss the flood initiation (section 5.1), the tremor generation during the flood propagation in the context of the known flood path (section 5.2), the tremor generation in the cauldron area in the context of the pressure drop (section 5.3), and the tremor from the river (section 5.4).

Finally, we put the propagation speed of the flood into a global context (section 5.5). Our paper will be a reference for further detection and classification of seismic signals associated with flood-related geophysical processes.

## 2   The Skaftá Cauldrons

Localized geothermal melting of ice at the base of a glacier causes the formation of a depression in the glacier surface, often surrounded by cylindrically symmetric crevasse patterns. The depression leads to a decrease in ice-overburden pressure and

consequently forces geothermal meltwater, geothermal fluids, percolating rain water, and surface meltwater to accumulate beneath the cauldron. Water accumulates as a subglacial lake until the water pressure is close to the ice-overburden pressure at the location of the weakest seal near the edge of the lake. The lake drains rapidly after outflow begins through the seal (Björnsson, 1977, 1988, 2003).

Two subglacial lakes in the western part of Vatnajökull ice cap, southeastern Iceland, 10 and 15 km WNW of Grímsvötn

volcano, are the source of regular jökulhlaups in the Skaftá river (Fig. 1) (Björnsson, 2003; Jóhannesson et al., 2007). The eastern and western cauldron each have a width of 1–3 km, depth of 50–150 m and host ca. 100 m deep subglacial lakes at their highest stage shortly before jökulhlaups are released (Jóhannesson et al., 2007). Combined, they drain approximately 65 km$^2$ of the ice cap (Pálsson et al., 2014). Finite-element ice-flow calculations show that ice-surface depressions caused by the emptying of the subglacial lakes are considerably larger than the footprint of the corresponding water body at the glacier bed (Einarsson et al., 2017).

Jökulhlaups from the Skaftá cauldrons into the Skaftá river catchment have been observed since 1955 (Zóphóníasson, 2002; Björnsson, 1977; Þórarinsson and Rist, 1955). Floods earlier than 1955 may have taken alternate drainage pathways outside the ice margin into Langisjór lake without leading to noticeable floods in the river course further downstream (Björnsson, 1977; Tómasson and Vilmundardóttir, 1967). The jökulhlaups occur every 1 to 5 years from each cauldron with volumes of 0.05 to 0.4 km$^3$ and maximum discharge rates of 50 to 3000 m$^3$/s (Björnsson, 1977, 1992; Zóphóníasson, 2002). The locations of the flood paths were estimated based on the gradient of the hydraulic potential derived from radio-echo sounding studies of the ice thickness and bedrock topography beneath the ice (Björnsson, 1986, 1988). The geometry of the hydraulic potential based on the radio-echo sounding measurements and the observed outlet locations of jökulhlaups at the ice margin of Skaftárjökull outlet glacier indicate that the location of the subglacial flood path shown in Fig. 1 has remained the same since jökulhlaups in Skaftá were first reported in the 1950s.

## 3 Methods

### 3.1 Seismic Network and Seismic Array

The seismic signals generated by the flood were recorded using two seismic arrays (clusters of seismometers) operated by the Dublin Institute for Advanced Studies (DIAS) with an aperture of 1.6 km (black inverted triangles in Fig. 1) and the Icelandic Meteorological Office's (IMO) national seismic network, SIL (Böðvarsson et al., 1996; Böðvarsson and Lund, 2003) (black triangles in Fig. 1).

In the Vatnajökull region, the SIL network consists of broadband (Güralp 3ESPC 60s and 120s, Güralp 6T Flutes 10s, Trillium Compact 20s, Geotech KS2000 100s) and short period (Lennartz 5s) stations, some installed on nunataks and within the ice. This network monitored the seismic activity associated with the flood. The closest stations DJK, HAM and GRF at 13–15 km distance from the draining cauldron recorded both quakes and tremor, but event locations, which mostly relied on P- and S-wave arrivals on the 3 to 7 closest stations, were hampered by emergent onsets, small magnitudes and low signal/noise ratios as well as station failures (HUS and horizontal components of DJK) and frequent earthquakes from Bárðarbunga volcano following its eruption in 2014/15 (Sigmundsson et al., 2014). As a result only a few events were located by the automatic SIL system. A review of the seismic network records, as part of this study, has enabled location and magnitude determination of 45 events near the cauldron and the flood path (Fig. 1) between 28 September and 2 October, using the SIL analysis software and velocity model (Rögnvaldsson and Slunga, 1993; Stefansson et al., 1993). Hypocentral location uncertainties are, however, rather large, up to 2–4 km in some cases.

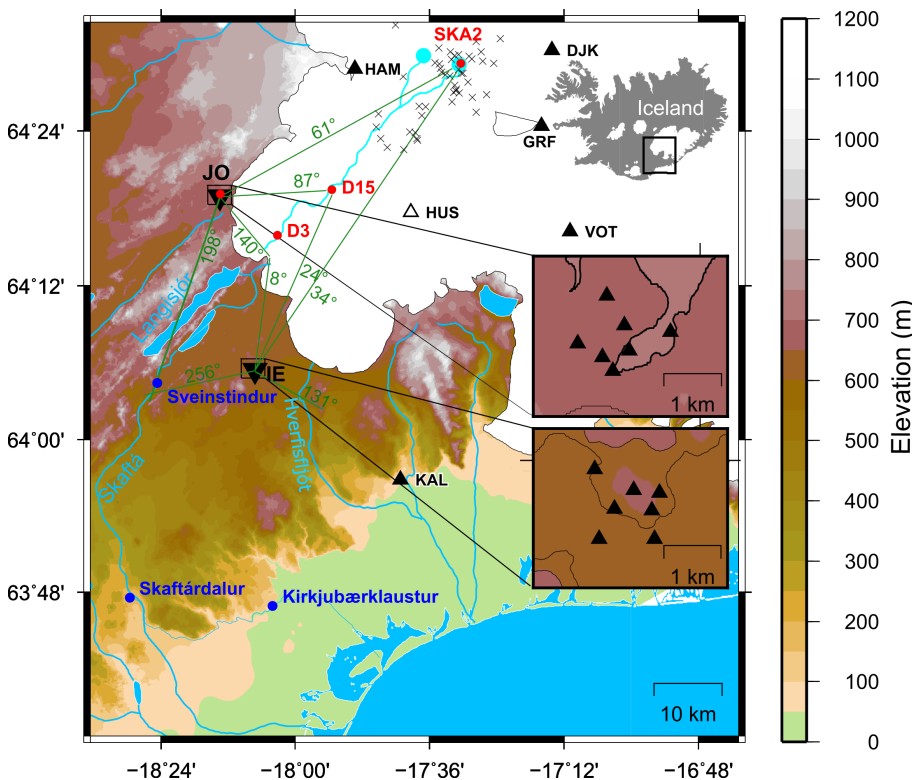

**Figure 1.** Instrument network on and west of Vatnajökull ice cap monitoring the 2015 jökulhlaup in Skaftá river. The eastern and western cauldrons (cyan dots) and subglacial and subaerial flood path (cyan line) (Björnsson, 1986, 1988) are marked. The locations of Jökulheimar (JO) and Innri-Eyrar (IE) seismic arrays (inverted black triangles), single seismic stations (black triangles, open if not working during the jökulhlaup), the GPS instruments (red dots) and hydrological stations (blue dots) are indicated. Locations of quakes are marked with black ×-signs and back azimuths from JO and IE with green lines with numbers. Note the two back azimuths from IE array that point to Svartifoss waterfall in the river Hverfisfljót to the east and rapids in Skaftá river near Sveinstindur to the west. The insets show (top) an overview of Iceland with glaciers marked in white, (middle) geometry of the JO array and (bottom) geometry of the IE array.

The telemetered arrays, specifically installed to record subglacial floods, consist of seven stations (six 3-component Güralp 6TDs (30 s to 100 Hz) and one 3-component Güralp 3ESPCD (60 s to 50 Hz)) at Jökulheimar (JO, centered at the SIL station
JOK) and Innri-Eyrar (IE, centered at the SIL station IEY) southwest of Vatnajökull at 38–52 km distance from the cauldron (5L seismic network, (Bean and Vogfjörd, 2020)). During a flood, these arrays can be used to locate and track the emergent, long-lasting, low amplitude tremor. As preprocessing steps we detrend, instrument-correct and down-sample the recordings to 20 Hz and divide them into 1 h long time windows. At the JO array, local noise sources dominate at 1 Hz while the flood–related signal was strongest around 1.3 Hz and clearly detectable above the noise. Our analysis and interpretation here is mainly based
on the results from one array (JO) and the *a priori* knowledge of the subglacial flood path.

### 3.1.1 Array Data Analysis in the Time and Frequency Domain

**FK-analysis in the frequency domain**

We performed frequency–wavenumber analysis on the array data, using the vertical component of the signal filtered between 1.2 and 2.6 Hz with a moving 18 s long time window as described in Capon (2009) and implemented in Beyreuther et al. (2010) and Megies et al. (2011). We repeated the analysis using only the North and only the East components to assess the dominance of the river noise. We did not rotate the horizontal components to radial and transverse before the analysis since we face a migrating source. As a consequence we here only discuss and show the derived back azimuths for the horizontal components but not the slownesses. A grid search for maximum power was carried out in a horizontal slowness grid with a stepsize of 0.02 s/km and limits of $\pm 1.0$ s/km. The resulting time series contain back azimuth and slowness, which reflect the direction and steepness of the dominant incoming wave, respectively. Time windows with a semblance of less than 0.3 were discarded. The semblance is defined as the ratio of the coherent energy to the total energy in the waveform stack within the time windows of analysis (e.g. Kennett (2000)).

The uncertainty in the back azimuth and slowness estimates based on the array geometry, is estimated using the width of the peak in the array response function in the horizontal slowness grid: We determine back azimuth and slownesses of all points that have a power of at least 95% of the maximum (Eibl et al., 2017a). We discard points with an uncertainty in back azimuth greater than 12° and uncertainty in slowness greater than 0.2 s/km. The resulting mean uncertainty for each back azimuth or slowness estimate was 4.2° and 3.0° in back azimuth and 0.04 s/km and 0.03 s/km in slowness at JO and IE array, respectively. When binning back azimuth estimates over a longer time period it becomes apparent whether the tremor is emitted at a spatially confined location, as indicated by little variation in the back azimuth with time, or whether the tremor source comprises a larger region, as indicated by scattering in the back azimuth with time.

**Beam stacking in the time domain**

In addition, we performed beam stacking in the time domain as an alternative to FK-analysis in a frequency band from 5 to 20 Hz using a moving time window of 0.5 s. Though similar to FK-analysis, it is more efficient for very short time windows. The same settings were used for the slowness grid and the required minimum semblance. We also discard time steps where the corresponding absolute slowness was above 0.3 s/km. Though not strictly necessary for stable results, the latter limit was introduced to exclude parameter ranges where the array aperture may be too large for the analyzed frequency range. This second type of processing was introduced after visual inspection of the waveforms had shown that certain transient signals were too short and of too high frequency to be detected with the lower frequency FK-analysis targeting the flood-related tremor sources. Most of these events are not visible on the SIL stations. These events are further discussed in the context of the tremor generation.

### 3.1.2 Root Median Square Amplitude

To additionally assess the tremor amplitude, we calculated the Root Median Square (RMeS) of the seismic recordings: The data were instrument-corrected, detrended, tapered and filtered between 1.0 and 2.0 Hz. The vertical component of the velocity

seismogram of one seismometer in each array was divided into 60-minute long time windows and RMeS was calculated. The time window was then shifted, allowing 75% overlap. Calculating RMeS instead of RMS strengthens the tremor signal while giving less weight to shorter events such as earthquakes (Eibl et al., 2017a). To assess the spectral characteristics of the seismograms and enhance the continuous signals in the tremor, spectrograms were created using window lengths of 64 to 256 s and overlap of 50–70%. We calculate spectra of up to 45 min long time windows. For this we split the data into 1 min long non-overlapping time windows, calculate the spectrum for each time window and stack the resulting spectra to enhance dominant frequencies.

### 3.1.3 Template Matching

To find additional signals of distinct and possibly repeating earth- or icequakes, we first used a STA/LTA-based event detector implemented in Snuffler (Heimann et al., 2017) to find template events. This STA/LTA-based detector calculates the ratio of a short time average window (STA) and a long time average window (LTA) (Allen, 1982). It was set up with a short window of 30 s and a long window of 90 s, centered over the short window. We applied it to the 3-component waveforms of the JO and IE arrays, filtered in the frequency range of 1 to 15 Hz. We normalized the STA/LTA traces for each component separately, then averaged them and used peak detection with a threshold of 0.3 to define detections. With this procedure we found 615 events between 28 September and 5 October 2015 and used these as templates in the following cross-correlation search.

For each template, we cut out time windows of 60 s and calculated cross-correlations (with a gliding normalization) for all components at all stations of the IE and JO arrays to find similar, weaker events. Both, template and continuous waveforms were filtered between 1 and 15 Hz. The continuous cross-correlation signals (between template and continuous waveform) of the individual stations and components were stacked and normalized by the number of contributing stations and components in each processing time window separately. Detections were defined where the stacked and normalized cross-correlation value exceeded a value of 0.2. Higher cross correlation values are not reached because both template and matched event are noisy, the noise levels at some stations are higher and the length of the template is longer than the event's signal. The average background level of the cross-correlation of the event templates with the continuous waveforms is on the order of 0.015. Against this, a value of 0.2 is highly significant and only reached when signals of both events arrive at all contributing stations and components without any differential time delays and also only when the coda of both events is qualitatively very similar (Suppl. Fig. A1). The resulting catalog consisted of 669 events. While the template matching produced only about 9% more detections than the STA/LTA-based detector, we noticed a significant number of cross-detections between the templates. These were further analyzed by event-wise cross-correlation and cluster analysis.

### 3.1.4 Event Clustering

To find potential clusters of seismicity, we analyzed the waveform similarity of the 669 events from the template matching catalog. In our first attempt we used the waveforms at JO array for the clustering. However, the waveforms were too similar and hence only sorted into one cluster. We tried several stations and finally present the results from station HAM, as this station is close to most of the sources, contains little noise and shows the highest waveform variability for the events in the catalog

(section 3.1.3). This was the seismic instrument closest to the flood path where all three components were operational over the time interval of our analysis. We cut time windows of 60 s around each event, filtered the waveforms to the frequency range 1 to 6 Hz and calculated pair-wise cross-correlations, keeping their maximum values and time lags for all event combinations. The cross-correlation maximum values were used to form a precomputed distance matrix for the OPTICS clustering algorithm (Ankerst et al., 1999), as implemented in the scikit-learn package (Pedregosa et al., 2011). We used $(1-C)^p$ with $p=8$ as the measure of distance, where $C$ is the normalized cross-correlation maximum of an event pair. 203 events were clustered into 20 different clusters and the remaining 466 events were left unclustered by the algorithm. The exact number of clusters formed and the fraction of events which were put into the clusters depended on the value of $p$ and on the tuning parameters of the OPTICS algorithm but the overall picture obtained was robust, which we verified by visual inspection of the waveforms in each cluster. For further inspection, we extracted back azimuth and slowness at JO array from the already computed list of back azimuths used for the tremor study (section 3.1.1).

## 3.2 GPS and Hydrological Measurements

A streaming Trimble NetRS GPS instrument was installed by the IMO near the center of the eastern cauldron (SKA2) from July 2014 to October 2015 for early-warning purposes. Two additional identical GPS instruments were installed above the subglacial flood path on the glacier at 15 and 3 km distance from the ice terminus (D15 and D3, respectively, Fig. 1 and Einarsson et al. (2016)). Mainly D15 was used to constrain the travel time of the subglacial jökulhlaup. Instrument D3 was washed away when a small part of the flood water hydrofractured through the ice. However, it was found the following summer on the surface of the glacier. Data were recovered from the internal memory card and used to constrain the speed of the subglacial flood wave, although the data do not contain information about the movement of the ice during the flood at this location. The GPS data were processed using the GAMIT-Track utility (Herring et al., 2015) with the continuous GPS station at JOKU in Jökulheimar as base.

Hydrological data of the Skaftá river were obtained from IMO's pressure-sensor stage meters at Sveinstindur, 28 km downstream from the glacier margin, and Skaftárdalur, 40 km downstream from Sveinstindur (Fig. 1) and include the river level, electric conductivity and water temperature. The river level was used to calculate flood discharge using a rating curve.

## 4 Results and Interpretation

### 4.1 Propagation of the Subglacial Flood According to GPS and Hydrological Measurements

Based on the GPS and hydrological instruments, the flood started to propagate in the early hours of 30 September, reached D15 at 17:30 of the same day and the hydrological station 25 km downstream at 4:00 on 1 October. However, the GPS instrument SKA2 already recorded a slow subsidence from noon on 27 September. Slowly increasing outflow of water from the subglacial lake on the order of a few cubic meters per second started at this time but the water was stored subglacially near the cauldron for about 3 days (Fig. 2a).

The subglacial lake emptied rapidly and the 300 m thick ice-shelf dropped by approximately 60 m in 24 hours. The subsidence of SKA2 on top of the ice-shelf slowed down rather abruptly at a lowering of about 66 m, accelerated again and subsided by another 17 m by 3 October (Fig. 6a). It should be noted that at the end of the subsidence the GPS instrument was not located at the deepest part of the cauldron. Mapping of the ice-surface geometry of the cauldron after the jökulhlaup showed two deep pits lowered by close to 40 m more than that at the center, probably caused by local maxima in the geothermal heat flux from geothermal vents at the glacier bottom, which melt large domes into the bottom of the ice-shelf. The continued slower subsidence recorded by the GPS instrument may have been caused by the local collapse of such thinner areas of the ice-shelf.

The flood started to lift the overlying ice at D15 for one day, reaching a maximum of approximately 1 m at 16:00 on 1 October (Fig. 2a). The flood was accompanied by an order of magnitude increase in horizontal velocity of the glacier to 1–2 m/d. The glacier surface then subsided over a two day period back to the level before the jökulhlaup. While the subsidence in the cauldron hovered around 66 m, the flood lifted D15 to its maximum height.

While the deformation at D15 was high, we detected 40–100 high-frequency transient signals per hour featuring semblance values of more than 0.35 on JO array from the lowermost 15 km of the glacier (Suppl. Fig. A3). The background rate before the flood arrived at D15 is 0–10 transients per hour (Table 1). Some events featured clear body wave arrivals. Our detector, however, did not find more of these clear events but found instead coherent body wave energy that closely followed the flood front. Interestingly, the initial water front was followed by seismicity from 90 to 120° between 00:00 and 03:30 on 1 October. At the same time the detected Type 1 tremor was strongest at the terminus. During the largest lifting of the ice sheet, the transient seismicity on the lowermost 15 km died down.

The D3 instrument detected no changes in glacier motion before midnight on 30 September. The flood must, therefore, have arrived later at this location, 3 km from the ice terminus. As river stage and conductivity rapidly increased from 04:00 on 1 October (Fig. 2b), the flood front most likely reached the ice terminus between 01:00 and 02:00 and fractured the ice at several locations 1–3 km from the terminus. These outbreaks were marked by up to 3–5 m high and 10 m wide ice fragments on the surface of the glacier, alongside debris deposited from the floodwater. Towards the ice terminus the fragments decreased in size, down to a few centimeters or tens of centimeters in diameter near the ice margin. After the first pulse, water continued to flow beneath the ice to outlets at the margin. The transient, high-frequency events support an arrival at the ice terminus shortly after midnight on 1 October.

### 4.2 Flood Initiation and Quakes Recorded on Closest Seismic Stations (SIL)

Seismic records from HAM and DJK, the stations closest to the draining cauldron, show that the onset of cauldron subsidence and slow outflow of water on 27 September did not immediately trigger seismic activity in the cauldron area. The first located quake occurs 16 hours later in the early hours of 28 September, within 4 km distance west of the cauldron center (Table 1, Fig. 1 and 2c). Only two events of M 0.7 were located that day in the cauldron area and another two smaller events in the afternoon and evening of the following day. On 30 September when the cauldron subsidence started accelerating, the seismic activity also escalated, to well over 100 events/hour in the afternoon (Fig. 2c to e).

However, because of the small magnitudes and overlapping signals, only 12 events could be located. Five of them were south-

| Term | Definition |
| --- | --- |
| Quake | 45 events located using the SIL network. Possibly 22 icequakes and 23 earthquakes. |
| Icequake | 30% of 669 events clustered into 20 families. Back azimuth pointing to cauldron area. Likely due to ice-shelf collapse. |
| Transient | Distinct events migrating with the flood front. Possibly icequakes. |
| Tremor type 1 | Continuous signal migrating with the flood front from the cauldron to terminus. (JO: 60 to 140°) <br> Spectrum diffuse |
| Tremor type 2 | Back azimuth pointing to cauldron area (JO: 53.1±15.6°, IE: 50.1±16.6°) <br> 2.8 to 36.6-minutes-long bursts |
| Tremor type 3 | Back azimuth pointing to cauldron area (JO: 53.1±15.6°, IE: 50.1±16.6°) <br> 10-minute and up to 6-hour-long, harmonic spectral content |
| Tremor type 4 | Back azimuth pointing to rapids in Skaftá river (IE: 256°) <br> 2.5 day long continuous signal |
| Noise 1 | Back azimuth pointing to Svartifoss waterfall in Hverfisfljót. Unrelated to the jökulhlaup. (IE: 131°) |

**Table 1.** Overview and definition of terms used in manuscript.

west of the cauldron area, along or in the vicinity of the subglacial flood path. The location of the first such event, 13 km from the cauldron at 11:03 on 30 September supports the inference from the GPS and hydrological data that the flood front had started propagating from the cauldron area around 04:00 in the morning. The high seismic activity continued into 1 October,
when the cauldron subsidence was the fastest, until around 14:00 when both seismicity and rate of subsidence swiftly decreased. In this period, another 19 events could be located, of which 7 were along or near the flood path. As the seismicity decreased, signs of tremor bursts appeared – the first one at 12:15 on 1 October – and continued through 2 October when they were the dominant activity on the seismic records. During this final phase, 10 additional events were located in the cauldron area and two along or near the flood path.

A total of 45 seismic events were located in the cauldron area, mostly within 4 km distance from the center, and along or near the upper part of the subglacial flood path (Fig. 1). Their source depths locate predominantly near the surface, with 33 events within 1 km from the surface. Event magnitudes range from ML 0 to ML 1.6. No events were located along the lower half of the subglacial flood path. Due to the emergent event onsets and small signal/noise ratios, the event locations have considerable uncertainty. However, they mostly cluster within the expected source areas and the recorded signals show the expected charac-
teristics of shallow source depth and propagation in the near surface, which result in long P and S wavetrains and dominance of low frequencies (Vogfjörd and Langston, 1996) (Suppl. Fig. A2). The frequency content above background is from 1–8 Hz for the smallest events and 1–15 Hz for the largest (Fig. 4a and b). The duration of signals on the array stations is around 30–35 seconds (Fig. 4c). Dominant frequencies are at 1–3 Hz.

39 of those 45 events are also detected through our STA/LTA and template matching (see section 3.1.3) and 22 of them are
280 sorted into a cluster. We hence interpret the 22 quakes that are sorted into a cluster as icequakes, and the remaining ones as

earthquakes in the shallow crust.

We used these 45 quakes to test our arrays for a systematic bias in back azimuth for signals coming from that direction and depth. The source locations were used to calculate expected back azimuths at the JO and IE arrays. Performing FK-analysis on these events we could compare expected back azimuths and array–derived back azimuths. Due to a low signal-to-noise ratio, only 17 events could be used at the JO array and 12 events at the IE array. For these quakes, the array–determined back azimuths were on average 9.3° too low at the JO and 10.1° too high at the IE array. It is likely that the back azimuths are systematically shifted due to heterogeneities in the seismic velocity structure under W-Vatnajökull related to volcanic ridges beneath the ice cap striking north-northeast–south-southwestwards. We keep this in mind when discussing potential tremor source locations as this systematic shift may be expected to affect tremor sources as well if they originate in the same region. Squinting arrays with systematic offsets between the actual source and array back azimuths were reported in (Eibl et al., 2017b; Krueger and Weber, 1992; Schweitzer, 2001). Due to the short signal duration compared to the length of the moving time window in the array analysis, the events do not affect the calculated back azimuth of the tremor.

## 4.3 Seismic Tremor Related to the Advancing Subglacial Flood Front (Type 1)

We subdivide the seismic tremor into four types based on the back azimuth direction and possible migration with time, the spectral content, amplitude strength and evolution with time (Table 1). When the subglacial flood started to propagate from the cauldron, it was accompanied by a migrating tremor source (Type 1) and transient high frequency events that we only detected at JO array (Fig. 2f). While the flood front arrived at D15 at ca. 17:30 on 30 September (Fig. 2a), stronger Type 1 tremor started at 18:40 in an area about 1 km farther downstream. D15 reached its maximum elevation about 20 h later, while the strongest tremor came from a location farther south and became weak on the last 8 km of the subglacial flood path. Along the whole flood path, Type 1 tremor was strongest in regions of adverse bedrock slope (up-slope flow about 35 kilometers down the flood path) and close to the ice terminus (Fig. 2a and g). The back azimuth of the Type 1 tremor and the high frequency transients first reached 140° at 01:30 on 1 October. This strong tremor close to the ice terminus might be linked to the hydrofracturing of the ice that happened about 3 km upstream from the ice terminus. However, the tremor near the terminus was also exceptionally strong from 01:30 to 02:45 and 03:35 to 04:09 on 1 October (Fig. 2g). In general, we can conclude that Type 1 tremor was generated in an area spanning up to 20°, as measured from JO array, which migrated downglacier with time.

The black line in Fig. 2g shows the back azimuth corresponding to a point migrating along the flood path at 2 km/h that crosses D15 at 17:30 on 30 September. It aids the visual interpretation of the tremor back azimuths, as this curve shows how the location and shape of the flood path map into changes of the back azimuth with time. It indicates that most of the tremor back azimuth is consistent with source locations upstream of a migrating flood front with a rather constant propagation speed near 2 km/h. Tremor is further generated in an elongated area as tremor is visible for up to 12 hours at some locations.

## 4.4 Icequakes originating from the Cauldron Area

The 669 seismic events were visible both on JO array and the stations from the SIL network near the cauldron (Fig. 5a and b). Back azimuths derived from the JO array indicate that most of these events originated in the cauldron area. Waveform-similarity-based clustering at station HAM (section 3.1.4) revealed that about 30% of these could be grouped into up to about 20 families of highly similar events (Fig. 5c). Visual inspection of the waveforms also shows high similarity between the families. This type of seismicity started at 23:24 on 29 September and ended at 12:38 on 3 October, with high activity occurring between 08:29 on 30 September and 04:45 on 1 October. The clusters were active for different time spans but with significant overlap. For example the cluster shown in blue in Fig. 5 was active over the entire time span of high activity.

The remaining, unclustered events mainly fall into 3 categories: a major part showed qualitatively similar waveforms in frequency content, duration and waveform pattern as the clustered events when inspected visually, but were possibly too weak or too noisy to be picked up by the clustering algorithm. A second part could be attributed to bursts of high energy within the Type 2 and 3 tremor. The remaining were mostly unrelated regional events not originating from the cauldron or glacier.

In the frequency range analyzed with this method, no distinct events originating from the flood propagation path could be identified with certainty. The back azimuths associated with the clustered events is 50 to 70° from JO array, while the slowness is in the range of 0.7 to 0.9 s/km. The events of all clusters lie in this range without any clear pattern. This is in accordance with the similar waveforms that we noticed across clusters. Note, that the back azimuth shown in Fig. 5 may be incorrect for some events due to simultaneous arrival of Type 1 tremor in the processing time window of the array analysis.

## 4.5 Subglacial Episodic Tremor from the Cauldron (Type 2 and Type 3)

At a later stage of the flood the tremor character changed and became stronger and episodic. We hence detected it on the JO and IE array (Fig. 6g and h). On the closest stations from the SIL network (e.g. GRF), a tremor band around 2 Hz was detected from 09:30 on 1 October. A first weak tremor episode was detected at noon with about eleven episodes in total (Fig. 6). For comparison, the subglacial flood from the Western Skaftá cauldron in January 2014 was followed by 7 episodes (Eibl et al., 2020). Each episodic tremor event consisted of two distinct tremor types that we refer to as Type 2 and Type 3 in the following

(Fig. 7a to c). Note that Eibl et al. (2020) do not separate these types but call them 'Type 2 tremor'. The episodic tremor stopped at 22:00 on 2 October when the ice-shelf had almost fully subsided. The first and last episodic tremors were weaker while the strongest and longest started at 01:11, 11:03 and 13:47 on 2 October, before the last phase in the settling of the ice-shelf started at 14:00 (Fig. 6a, f and h). This last settling is characterized by an exponential decrease in the ice-shelf height.

Each episodic tremor started with an emergent, 2.8 to 36.6-minutes-long burst (Fig. 8a to h) with a frequency content up to 7 Hz. We refer to it here as Type 2 tremor. It features distinct but not equal-spaced frequency bands from 0.8 to 1.8 Hz (Fig. 7d and e). The dominant frequency changes with time and is within the range of 0.85 and 1.2 Hz (Fig. 8i to p). One has to keep in mind though that other subglacial (Type 1) or subaerial (Type 4) tremor sources might influence the spectra specifically when the Type 2 tremor bursts are weaker.

Type 2 tremor bursts were followed by a 10-minute and up to 6-hour-long harmonic tail (Fig. 8i to p) with several distinct frequency bands in the range of 0.8 to 1.6 Hz. We refer to this tail as Type 3 tremor, which has an overtone spacing of about 0.3 Hz (Fig. 7f and g). Similar to Type 2 tremor, the peak frequency of Type 3 tremor changes slightly with time but is mostly around 1 Hz (Fig. 8y to af). The fundamental frequency is in both cases not visible due to low-frequency noise.

We isolated the back azimuths of Type 2 and Type 3 tremor (Fig. 8ag and ah) and created histograms with 2° wide bins (Fig. 8ai and aj). The gray bars in Fig. 8ag and ah indicate the uncertainty associated with each back azimuth estimate, based on the array geometry (see Methods). We do not see a clear difference in back azimuth or slowness when comparing Type 2 and Type 3 tremor. The median back azimuth $\pm$ standard deviation during all tremor bursts were 53.1$\pm$15.6° at JO and 50.1$\pm$16.6° at IE. The large standard deviation/ scatter of these values indicates that Type 2 and 3 tremor was generated over a wide region.

Given that slownesses indicate a Type 2 and 3 tremor source in the bedrock and that back azimuths roughly point towards the eastern cauldron, they might be generated in roughly the same region as the icequakes. For these icequakes, we determined back azimuths that are 10.1° too large at IE and 9.3° too low at JO array. If the tremor back azimuths are affected similarly, Type 2 and 3 tremor should in reality be generated at an average back azimuth of 62.4° from JO and 40.0° from IE. This corresponds approximately to the area of the cauldron (see Fig. 1).

## 4.6 Subaerial Seismic Tremor and Noise from the River

At the JO array, we sometimes detected seismic signals associated with the subaerial flow in the nearby glacial river. Note that we only detected this signal at times when the tremor from the glacier was weak e.g. in the early hours of 30 September, when the flood just started. At the final stage of the flood, the subglacial water entered the glacial river Skaftá and drained towards the sea. The increased volume of water in the river generated Type 4 tremor. The back azimuths from around 198° as seen from the JO array can be associated with rapids near Sveinstindur that are at 29.5 km distance from JO array (Fig. 1). These were visible in the derived back azimuths on 1 and 2 October when processing the vertical component at the JO array (Type 4 tremor in Fig. 2g and Fig. 6g). On the horizontal components this noise is not detected above the noise floor.

The back azimuths derived using the IE array show a continuous dominant noise source at 131° (Noise 1 in Fig. 6g). This noise is strong, detected on all three components and masks most tremor sources. Nevertheless, the Type 4 tremor source at 256° from IE array was detected on the vertical components from 19:45 on 1 October when the river stage at Sveinstindur reached 7 m (Type 4 tremor in Fig. 6g). The Type 4 tremor reached its largest amplitude at 0:30 on 2 October when the river stage reached its maximum of 7.75 m. Finally, the Type 4 tremor was not detected on the vertical components after 16:00 on 3 October when the river stage at Sveinstindur had dropped back down to 6.2 m height and continued to decrease thereafter. On the horizontal components the Type 4 tremor is visible from 7:45 on 1 October when the river stage had increased - 1.8 m above the normal flow rate - to 4.3 m height (Fig. 9). This tremor source faded once the river stage dropped to 3 m height and hence dominated the seismic wavefield on the horizontal components longer than on the vertical components.

## 5 Discussion

### 5.1 Triggering of the Flood

The cauldron subsidence, as recorded by the GPS instrument on top of the subglacial lake, started on 27 September 2015. The first quake appeared about 16 h later and seismic activity peaked when the cauldron subsidence sped up and the flood propagation started. Therefore, no seismic activity in the form of earthquakes or icequakes was detected when the cauldron seal failed. This failure led to an initial slow water loss from the subglacial lake, which only a few days later developed into a flood wave migrating downhill.

In contrast, a flood from a glacier-dammed, marginal lake on Glacier de la Plaine Morte, Switzerland in 2016 was presumably triggered by icequakes (Lindner et al., 2020). When the melt season started and progressed, Lindner et al. (2020) were able to track how an efficient draining system progressed upglacier. Since they recorded icequake signals near the lake basin in the 24 h period before the lake drained, they suggest that hydrofracturing linked the ice-dammed marginal lake to this drainage system and led to a flood. Here, at the Skaftá cauldrons such hydrofracturing might not be a relevant flood trigger, since the lake is located at the bedrock-ice interface and therefore does not cause vertical hydrostatic pressure aiding a hydrofracturing event.

The quakes recorded here are interpreted as a sign of brittle failure. They are located near the bedrock-ice interface. They appear

once the water started to slowly migrate from the subglacial lake and might therefore be caused by the pressure change induced in the bedrock. Alternatively these quakes could be signs of further failure of the cauldron seal, or linked to the subsidence of the ice layer, opening crevasses that are visible on the surface (Fig. 10). These quakes, potentially caused by crevasses, are clearly distinguishable from the various tremor sources due to a higher frequency content.

Based on the results of our waveform similarity analysis (section 3.1.3) the cauldron area produced a lot of seismic events which we interpret as icequakes from 8:29 on 30 September to 4:45 on 1 October. In this time period the largest volume of water drained from the subglacial lake and the ice-shelf subsided by more than 30 m reaching its fastest subsidence rate. Our clustering revealed that the waveforms of these events can be clustered into 20 families. Considering slip during the gradual collapse of the ice-shelf, happening at different times and at different locations near the up to 3 km-wide cauldron might cause slightly different waveforms that are sorted into different families. The character of these events could be explained by repeated stick-slip on the same or on close-by fault segments in the ice. The events might be repeating or occur in close proximity to each other. The shortest contributing wavelengths are on the order of about 130 m. Due to several event families and similarity of events within a family, it is more likely that these events are icequakes generated when parts of the ice-shelf collapse rather than earthquakes. The progressive activation of different clusters (Fig. 5) might reflect the gradual slip along different fault planes in the ice. Similarly, during the Bárðarbunga caldera collapse in 2014/15 earthquakes on the northern and southern ring fault segments were different and could be separated into two major families (Gudmundsson et al., 2016).

### 5.2 Flood-related Tremor Generation (Type 1)

For the Skaftá 2015 jökulhlaup, the subglacial Type 1 tremor source moved gradually southwards accompanying the subglacial propagation of the flood. Based on the geometry of the flood path, we calculated the back azimuths we would expect from a flood front propagating at constant speed along the path (see section 4.3 and Fig. 2). The back azimuths of the Type 1 tremor closely followed these expected back azimuths of the flood front during the flood. Tremor was furthermore generated in a wide migrating region and was strongest in regions with adverse bedrock slope (Fig. 10). As plausible tremor models we consider turbulent flow, impact during bedload transport, resonance and repeating icequakes.

Resonance might be triggered by water flow in a subglacial channel or between the bedrock and ice layer. However, if resonance is triggered in this manner we would expect tremor generation along the whole flood path, not mainly following the propagating flood front. In addition, the strong tremor in regions of adverse slope cannot be explained by this model.

If tremor were induced by turbulent flow, we would expect a high tremor amplitude along the entire flood path upstream of the flood front. Instead, we find that the tremor source moved and started for example at the GPS instrument D15 shortly after the first sign of the flood was detected there. The tremor amplitude is highest while the glacier is being lifted, especially in regions of adverse bedrock slope, and the tremor source moves southwards following the flood front. Additionally, we detected differences in tremor generation along the flood path with little tremor generation on the upper half of the flood path. As we expect similar water flow speeds and turbulence along the whole flood path upstream of the flood front, our observations are

not consistent with tremor generation by turbulent flow.

Our surface wave observation, however, is consistent with the Rayleigh wave-dominated glaciohydraulic tremor reported by Vore et al. (2019) at more than 1 km distance from the source. Lindner et al. (2020) recorded tremor before and during a flood at the Glacier de la Plaine Morte, Switzerland and interpret it as signs of ice fracturing, moulins and moulin resonance that hide potential tremor linked to turbulent flow. Lindner et al. (2020) did not manage to track the propagating flood front, but located a persistent tremor source near an outlet and interpret it as linked to subglacial water flow. This might be consistent with our observation that tremor is strongest near the outlet and on the lowermost part of the flood path. However, for Skaftá floods we were far from the source (>50 km) and it remains to be shown whether tremor signals along the flood path have varying amplitude or whether the signal amplitude was merely modulated by the distance between the flood and array location.

Tremor caused by bedload sediment transport is thought to be characterized by a frequency content of more than 9 Hz while pressure fluctuations in turbulent flow is thought to be characterized by a frequency content of less than 9 Hz (Bartholomaus et al., 2015; Gimbert et al., 2016). These observations are made a few kilometers from the source and at larger distances the high frequencies will be attenuated. At 10 to 52 km distance, we observe tremor that is strongest around 1.3 Hz. The bedload transport studies indicate that bedload sediment transport is at this distance an unlikely generating source in the present case. Additionally, the Skaftá floods might not transport much sediments in comparison to other sites globally, as mentioned by Bartholomaus et al. (2015); Gimbert et al. (2016); Cook et al. (2018).

We suggest that the Type 1 tremor was generated by the high strain rates caused by the advancing water front. The glacier is lifted quickly up to 1 m off the bedrock and hence behaves in an brittle way in contrast to its usual plastic behavior. The water front can flow into these newly formed cracks and propagate them further. This lifting is inferred to be typical for the front of fast-rising jökulhlaups (Jóhannesson, 2002; Björnsson, 2010; Einarsson et al., 2016, 2017; Magnússon et al., 2007). The area of increased velocity was studied with InSAR during a flood in 1995 and found to be at least 9 km wide (Fig. 6 in (Magnússon et al., 2007)). This mechanism implies that the ice underwent brittle fracturing that resulted in small, repeated, closely spaced icequakes that could merge into tremor as suggested by MacAyeal et al. (2008) for colliding icebergs.

Further evidence for repeated icequakes in the Type 1 tremor stems from the detected high-frequency events that closely follow the flood propagation. Interestingly, some areas apparently generate more Type 1 tremor i.e. around D15 or near the terminus, while other areas generate more high-frequency events i.e. the area between D15 and the ice terminus. This might be caused by normal or adverse bedrock slopes, the distance that possibly affected our event detection, or the bedrock roughness. Independently, high-frequency events are generated along the same path that we detect Type 1 tremor which in our opinion suggests a close link. These events are too weak to be clustered and the clustered events mainly focus on the cauldron area. This might be in line with tremor composed of icequakes where the templates vary in space and time as the flood front propagates and both the source and the path changes.

This process may be assumed to have been particularly intense at the tip of the flood front at each point in time but continued until the discharge reached the maximum at each location. According to this interpretation, the first strong tremor period around

02:00 on 1 October from the direction of the glacier terminus might be due to the hydrofracturing of the ice that is likely to have been especially intense as the flood lifted the thinner ice near the terminus. While most of the water continued to flow near the bedrock, tremor might have decreased after the hydrofracture reached the surface. The second stronger tremor period (around 04:00 on 1 October) most likely marks further ice fracturing as the flood discharge near the terminus increased. In addition, the

seismic sources generated in a lifted ice sheet might not couple well to the ground once the ice sheet is separated by the water layer.

Similarly, Behm et al. (2020) recorded a rapidly-rising jökulhlaup in Zackenberg river in Greenland that was accompanied by intense surface crevassing as inferred from seismic icequake detection by seismometers on the ice. They suggest that increased basal sliding leads to increasing seismicity and crevassing on the surface. We are most likely too far from the source to detect

these icequakes caused by the ice movement. However, we speculate that during the flood when parts of the ice cap speeds up, crevassing and seismicity intensifies. This might merge into what we record as non-harmonic tremor at more than 10 km distance due to scattering effects of the shallow bedrock layers (Ying et al., 2015). A GPR study in Greenland also detected basal crevassing (Behm et al., 2020), that might be similar to the hydrofracture we observed here, which here ruptured all the way to the ice surface near the terminus.

The migrating tremor source is spread over up to 20° along the flood path as seen from the JO array. This indicates that tremor does not start immediately when the flood front arrives but that the ice surface needs to be lifted further before tremor becomes visible from each location. In the context of tremor that is composed of icequakes, this suggests that a threshold number of icequakes are required before tremor is detected at our observational range. The width of the flood front, ~9 km as derived from InSAR studies (Magnússon et al., 2007), may also be expected to contribute to the spread of the calculated back azimuth.

The activated region is therefore large and icequakes are likely neither similar enough nor spaced regularly enough to generate harmonic tremor such as observed during strike-slip collisions of edges of icebergs MacAyeal et al. (2008). Large-scale sliding events of a glacier in Antarctica, not associated with a subglacial flood, were accompanied by tremor episodes at the ice–bed interface. They were interpreted as repeating earthquakes due to the clear presence of single events and harmonic character of the observed gliding tremor (Lipovsky and Dunham, 2016). Tremor in our case neither shows harmonic character nor clearly

repeating events that might compose it. The visible peaks in the seismogram of Fig. 2e are linked to the quakes around the eastern cauldron or other volcanically active regions. Additionally, our GPS recordings indicate that the glacier is lifted up to 1 m off the bedrock which led us to conclude that tremor might be generated by irregularly repeating icequakes while ice is hydrofractured rather than earthquakes on fault planes on the bedrock-ice interface.

We observe a substantial but short-lived increase in horizontal velocity of the glacier at each GPS location when the pressure

wave passes (Einarsson et al., 2016). The maximum of the velocity increase coincides with the maximum of lifting and decreases as the wave has passed by. The velocity increase is partly due to shear thinning but mostly due to increased basal sliding of the glacier. Both increased scraping at the bottom of the ice or stick-slip motion could lead to tremor generation. This would then follow the location of the flood front and would not be detected once the flood reaches the glacial river. This is consistent with the Type 1 tremor locations that stopped at the glacier terminus.

While other glacier-seismology studies (Bartholomaus et al., 2015; Lindner et al., 2020) report a tight correlation between discharge and tremor amplitude in the 1 to 10 Hz band, Eibl et al. (2020) do not report a correlating diurnal variation but rather a temporal offset in tremor amplitude and discharge. However, floods with larger peak discharge are still accompanied by larger tremor. The time offset might be caused by large distances between instruments or due to flow of water between the ice and bedrock in a wide area instead of a channelized flow (Eibl et al., 2020).

Nevertheless, we note that Type 1 tremor still continues to increase in amplitude after the flood front has reached the terminus of the glacier and has the highest amplitude early on 2 October for the 2015 flood. This tremor might be formed by the subglacial water flow. This interpretation is able to explain the following points: i) its magnitude follows the discharge on 1 October, ii) it seems to be generated along the whole flood path as seen by seismic signals that are not coming from only one specific direction, iii) it should be more distinct for the large flood from the eastern cauldron than for the small floods from the western cauldron as observed by Eibl et al. (2020).

If a GPS instrument measures a pronounced lifting of the ice this indicates that the capacity of the subglacial drainage system was overwhelmed by the abrupt water escape (Lindner et al., 2020). For example, Lindner et al. (2020) report a peak discharge and GPS/ ice lifting in the first hours when the moulin reached the lake bottom. Consequently, the GPS elevation lowered to the levels before the drainage, and the discharge measured in the river dropped, while the lake drained slower and incised into the ice to drain through the moulin. In this study, the GPS is lifted even more in a second pulse and the peak discharge is only measured 4 days after slow outflow from the lake was detected. The draining water overwhelmed the capacities of the subglacial drainage system leading to a pronounced ice lifting and widespread flow of water beneath it.

### 5.3 Cauldron Tremor Generation (Type 2 and Type 3)

Taking bias in the back azimuth from the JO and IE arrays into account, back azimuths during Type 2 and Type 3 tremor point towards an area near the eastern cauldron (compare Fig. 1 and Fig. 8ag to aj).

Each of these tremor episodes starts with an emergent burst and is followed by an hour-long harmonic tail. Similarly, Montanaro et al. (2016) reported 40 to 50 s long explosions with frequency content up to 4 Hz followed by a several minute long tail of elevated tremor during a flood from a semi-subaerial lake at Kverkfjöll, N-Vatnajökull, in 2013. They suggest explosions due to expansion of boiling fluid in the geothermal reservoir followed by vigorous boiling. Remnants of such explosions were observed from the air the following day. This observation was the first time such a subglacial tremor burst could be confirmed visually. While they reported a drop of 30 m at Kverkfjöll, we observed more than 100 m at the Skaftá cauldron and hence a larger pressure decrease that might cause the long tremor duration reported here.

The temperature in the subglacial geothermal area may be assumed to be close to the pressure boiling point of water except at shallow depths near the glacier bed (e.g. Gudmundsson and Björnsson (1991); Ármannsson (2016). This implies that a lowering of the overlying pressure by ~0.6–1 MPa, corresponding to a drop in water level of ~60 m as well as the lowering of the effective pressure in the lake due to bridging stresses in the subsiding ice shelf (Einarsson et al., 2017), will lead to a lowering of the pressure boiling point within the geothermal system in the range 5–15 K (Wagner et al., 2000). A lowering of the pressure boiling point of this magnitude will lead to vigorous hydrothermal boiling of water in shallow crustal rocks, which

explains the creation of body waves by this tremor source.

Given that the episodic tremor is very strong, we would like to discuss it in the context as a possible sign of a subglacial eruption. Based solely on the volcanic tremor recordings subglacial volcanic eruptions and explosions are difficult to distinguish. Within this context our geochemical water samples play a crucial role. Small volcanic eruptions are not likely to be the cause of Type 2 and Type 3 tremor, as water samples showed elevated concentrations of dissolved inorganic carbon (DIC) and major elements

including Ca, Mg, and B (see supplementary material for details on geochemical data). High DIC concentrations are indicative of sustained water–rock interaction prior to subaerial exposure, and peak concentrations during this flood are comparable to previous floods from the Skaftá cauldrons (Jones et al., 2015; Galeczka et al., 2015). Boron is a strong indicator of geothermal activity given its high mobility (Arnórsson and Andrésdóttir, 1995). The boron concentration peaked at 18.3 $\mu$mol/L in the pro-glacial river at Kirkjubæjarklaustur. This is over 4 times higher than measurements from the 2014 flood from the western Skaftá

cauldron at the same locality (Jones et al., 2015), indicating that the non-glacial-melt part of the floodwater is of geothermal origin that has taken years to accumulate.

The geothermal chemical signature of the water suggests that the reservoir built up gradually beneath the ice cap, with continued and long-lasting reactions with the bed rock. With no indication for eruptive activity beneath the ice, we suggest that Type 2 tremor reflects explosions while Type 3 tremor reflects boiling in the shallow crustal rocks (Fig. 10). The subtle changes in

the frequency content with time might reflect a changing environment that is confined by a first enlarging, then shrinking resonating void in the ice that is intermittently present when the water drained, and the ice has not settled yet on the ground. In subglacial environments, linking the observations of tremor presented here and the associated geochemical fingerprint is crucial. If in future only one of the two is available, our study will support a correct interpretation of the associated signals.

## 5.4 Tremor and Noise Generation by Rapids and Waterfalls, respectively

All three seismometer components in the IE array are significantly affected by a continuous noise source (Noise 1). It is likely that this noise source is caused by flow of water in a nearby river (Fig. 10). The Svartifoss waterfall located in a narrow gorge on the Hverfisfljót river 7 km southeast of the IE array is a likely noise source at 131°.

Near Sveinstindur there are strong rapids, which are at 15 km distance southwest of the IE array. This might generate the Type 4 tremor from 256°. On our arrays at more than 10 km distance from the Skaftá glacial river, we do not detect the flow of water

in the subaerial river apart from the locations with rapids. This is consistent with the suggestion by Gimbert et al. (2016) that at more than 1 km distance, the seismic signal caused by turbulent water flow dominates over the signal caused by bedload transport. The horizontal components of the seismometers in the arrays and consequently their back azimuth determinations are strongly sensitive to Type 4 tremor. This can help to track the amount of water in the rivers especially when processing the horizontal components. However, on the downside Type 4 tremor will also hide Type 1, 2 and 3 tremor linked to the subglacial

flood processes. In remote areas it is promising that subglacial floods also generate seismic signals in the subaerial, glacial rivers that can be detected by remote seismic networks.

We note the differences in slownesses between the signals generated by the rapids and those generated by the waterfall, that might reflect the different source-receiver distances. While the Svartifoss waterfall at 7 km distance might generate more noise

that is composed of surface waves, the rapids at 15 km might generate a mixed wavefield. Eibl et al. (2017b) similarly reported changes in slownesses that coincided with changes in the distance between the actively growing lava flow field and the array.

## 5.5 Speed of Floods Globally

Eibl et al. (2020) combined GPS, hydrological and seismic data to estimate the speed of the 2015 Skaftá cauldron flood. They derive an average speed of 2 km/h (0.6 m/s) for the lowermost 15 km of the flood path from GPS and hydrological measurements and an average speed of 1.4–2.4 km/h (0.4–0.7 m/s) along the flood path from seismic observations. Assuming constant speed upstream of D15, the timing derived for the start of the flood from near the cauldron at 04 on 30 September is in broad accordance with the time when the rate of cauldron subsidence started to accelerate.

We calculated the expected back azimuths along the known flood path and converted the distance along the flood path to time assuming a constant velocity of 2 km/h as discussed above (Fig. 2). The resulting change in back azimuth with time closely fits the initiation of Type 1 tremor observed at each location along the path, indicating (i) approximately constant flood propagation along the path and (ii) less bias in back azimuth for Type 1 tremor generated at or above the ice–rock interface along the flood path than Type 2 tremor generated at depth in the crust. We might expect less bias in the back azimuth for Type 1 tremor composed of surface waves traveling in the glacier ice, as ice is more homogeneous than volcanic bedrock. Whether these waves might be affected by the heterogeneous bedrock finally depends on the dominating wavelengths. The Type 2 tremor interpreted to be generated at depth in the crust is sensitive to possible heterogeneities in the seismic velocity structure of the crust. A strong Type 1 tremor source that we interpret as an area of substantial lifting somewhat upstream of the flood front clearly moved downglacier from an area near the cauldron to the glacier margin (Fig. 6). Exact estimates of the initial arrival of Type 1 tremor or the time of maximum tremor intensity at specific locations of the flood path are hard to derive because of the wide scatter in the data.

On the lowermost part of the flood path (D15 to the glacier margin) the average change in the back azimuth at the JO array may roughly correspond to flood front propagation of ∼15 km in 6–7 hours, in crude agreement with propagation estimated from GPS and hydrological data. The variation of the back azimuth from higher up the path to the glacier margin is harder to estimate because the Type 1 tremor signal from the upper part of the path is weak. It does not allow us to make a statement on how the propagation speed varies as the flood moves through the glacier.

The flood in September/ October 2015 propagated faster than other known jökulhlaups in Iceland. In October 1995, Magnússon et al. (2007) found that on the last 7 km of the subglacial flood path the speed of the flood front during a small jökulhlaup from the eastern cauldron was less than 0.06 m/s (0.2 km/h). This is an order of magnitude lower than the average speed we derive during the jökulhlaup in 2015, but Magnússon et al. (2007) described the flood in 1995 as an unusually small one, occurring unexpectedly only 3 months after a large jökulhlaup from the same cauldron. Einarsson et al. (2016) and Einarsson et al. (2017) report subglacial flood propagation speeds of 0.2–0.4, 0.1–0.3 and 0.4–0.6 m/s for jökulhlaups from the western cauldron in September 2006 and August 2008 and from the eastern cauldron in October 2008, respectively. Similarly, Eibl et al. (2020) report propagating speeds in the range of 0.2 and 0.4 m/s (0.9-1.6 km/h, 0.7-1.1 km/h and 0.8-1.3 km/h) for three subglacial floods from the western Skaftá cauldron. These differences might be caused by the flood size as Eibl et al. (2020) conclude that

floods with a smaller peak discharge propagate slower.

Larger velocities were derived on other glaciers in Iceland and worldwide. A large jökulhlaup from Grímsvötn, Iceland, in 1996 propagated at 5 km/h (Björnsson, 2003; Jóhannesson, 2002). Benediktsdóttir et al. (2021) calculated subglacial flood speeds during the Eyjafjallajökull eruption 2010 of 2.0, 2.5, 3.75 and 15 km/h for floods of similar size as we reported here. The propagation speed of subglacial floods in other glaciers worldwide have been reported as 6.1 m/s (22 km/h) for glaciers in the Pacific northwest (Richardson, 1968) and >2 m/s (>7.2 km/h) (Driedger and Fountain, 1989) at Mt. Rainier in Washington state in the US and >1.05 m/s (3.8 km/h) in Switzerland (Werder and Funk, 2009). Larger speeds might be due to larger flood sizes (Eibl et al., 2020), a larger gradient of the topography, path width or other factors that are not constrained in detail yet.

## 6  Conclusions

The September/October 2015 jökulhlaup in the Skaftá river was one of the largest measured floods in Iceland since the start of hydrological measurements at Sveinstindur in 1971. The flood was released after an unusually long interval of more than five years since the previous jökulhlaup from the eastern Skaftá cauldron. This subglacial flood was accompanied by a characteristic seismic sequence consisting of (i) a few-second-long icequakes generated by brittle failure during the gradual ice-shelf collapse above the subglacial lake. Then (ii) hour-long non-harmonic Type 1 tremor and high-frequency transient events follow the flood front from the cauldron to the ice terminus. We suggest that the source of the Type 1 tremor observed during the subglacial propagation of the Skaftá jökulhlaup is repeating icequakes generated while the ice is quickly forced upwards to allow water flow below. Consequently, (iii) approximately 10 to 15-minutes-long Type 2 tremor bursts occurred, followed by (iv) an up to 6-hour-long harmonic Type 3 tremor tail. Type 2 tremor bursts were interpreted as hydrothermal explosions in the cauldron area and and the Type 3 tremor as vigorous boiling. This episodic Type 2 and 3 tremors indicate the drainage of floodwater from the cauldron and the associated lowering of water pressure in the subglacial geothermal system, respectively. There is no indication of subglacial volcanic eruptions based on our geochemical water samples from Skaftá river. Finally, (v) we detected Type 4 tremor caused by rapids in the glacial river that strongly correlated with the discharge in that river. We hence also managed to detect and follow the flood as it traveled outside the ice and into the glacial river.

Globally most subglacial flood studies report on quakes (Behm et al., 2020) or tremor (Winberry et al., 2009; Bartholomaus et al., 2015; Lindner et al., 2020; Vore et al., 2019) during the floods. Iceland however seems to be a unique place where fast rising jökulhlaups may be followed by episodic tremor potentially caused by explosions and boiling in the the active geothermal systems driven by the active subglacial volcanic systems. Despite geothermal activity in Greenland or Antarctica (Fahnestock et al., 2001; Loose et al., 2018; Schroeder et al., 2014) a similar sequence of seismic signals remains to be reported in other regions with volcano-ice interaction. The methods described and knowledge gained here can aid in the identification of flood signals and their differentiation from eruption–related signals in other glacier-covered, volcanically active regions worldwide that can lead to hazardous flooding.

*Code and data availability.* Seismic data are available via GEOFON (5L seismic network, (Bean and Vogfjörd, 2020)). The FK-analyis was
performed using the freely available Python toolbox ObsPy. Event catalogs are available at GFZ Data Services (Eibl et al., 2023).

*Author contributions.* EE., CB. and KV. initiated the study conception and design. Data collection was performed and supported by BB., EE., BO., MR. and MJ. Data analysis was performed by EE., KV., BO., MJ., SH. and TD.. The first manuscript draft was written by EE. and all authors commented on previous versions of the manuscript. All authors read and approved the final manuscript.

*Competing interests.* There are no competing interests.

*Acknowledgements.* Field work, data collection and analysis was performed within the framework of FutureVolc, funded by the European Union's Seventh Programme for research, technological development and demonstration under grant agreement No. 308377. MTJ is supported by the Research Council of Norway (project numbers 223272 and 263000). We thank Bergur Einarsson for fruitful discussions, Martin Möllhoff, Heiko Buxel, Baldur Bergsson, Vilhjálmur S. Kjartansson and Þorsteinn Jónsson for technical support and Aoife Braiden for support in the field. Water samples were collected at Skaftárdalur and Kirkjubæjarklaustur by IMO staff. We acknowledge the assistance of Tómas Jóhannesson in providing data and contributing to the interpretation presented in the paper. We extend our thanks to Iwona Galeczka for processing of these samples at the University of Iceland.

The Landsvirkjun (National Power Company of Iceland) Research Fund, the Icelandic Road Administration, the Kvískerja fund and the Iceland Glaciological Society have supported the glaciological field work and research on jökulhlaups from the Skaftá cauldrons.

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

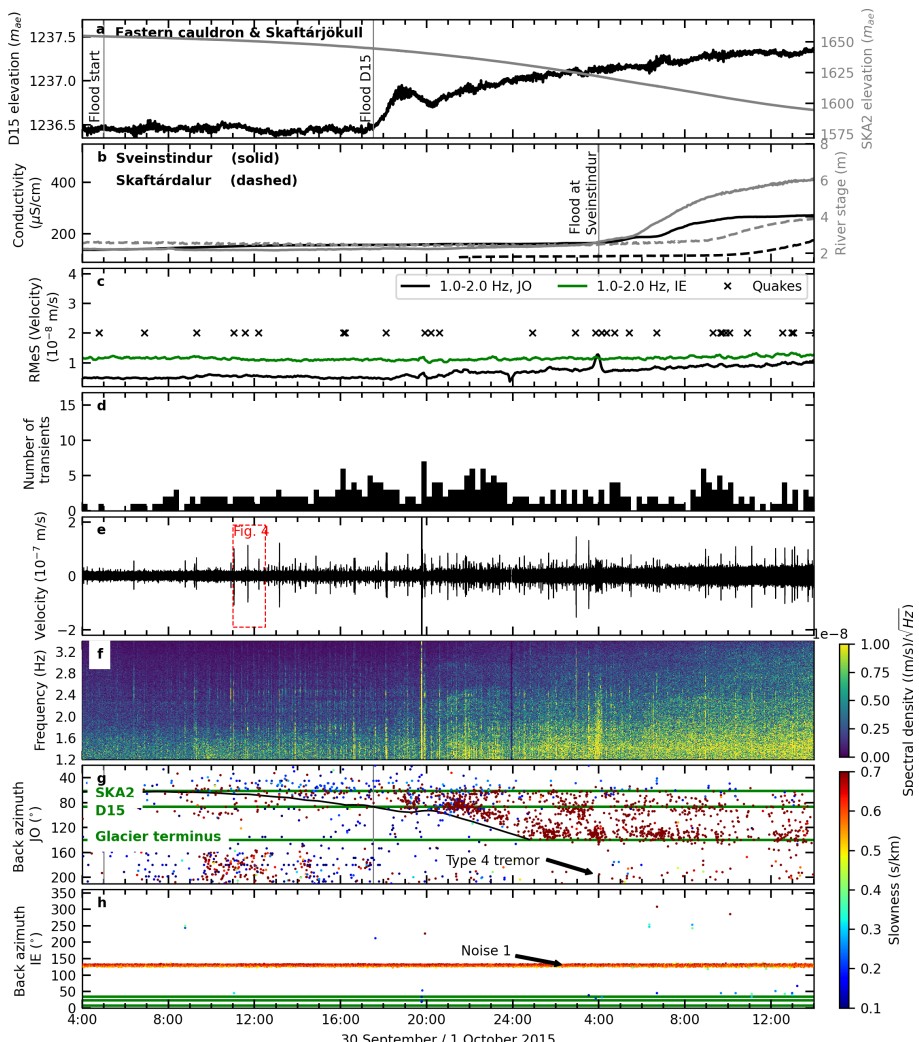

**Figure 2.** Subglacial Type 1 tremor followed the migrating flood front in September/October 2015 (for station locations see Fig. 1). Gray lines mark important events during the flood, detected by various instruments for comparison with the tremor. Subfigures a, b, e and f redrawn from Eibl et al. (2020). (a) The elevation of GPS instruments in the eastern cauldron (SKA2) and on Skaftárjökull (D15) above the GRS80/WGS84 ellipsoid (67.23 m at D15 and 67.72 m at SKA2). (b) River stage (gray) and electrical conductivity (black) of Skaftá river measured at Sveinstindur (solid) and in Skaftárdalur (dashed). (c) Root Median Square (RMeS) of the seismic amplitude filtered 1 to 2 Hz at JO and IE arrays. The occurrence times of located quakes (Fig. 1) are marked with black ×-signs. (d) Number of events detected with STA/LTA and template matching. (e) Vertical velocity seismogram from 04:00 on 30 September to 12:00 on 1 October 2015 filtered between 1.2 and 3.4 Hz. Some of the 'spikes' in the figure are not part of the tremor but short quakes from the cauldron area or earthquakes in other nearby locations like Bárðarbunga volcano (see Fig. 4). (f) Amplitude spectrogram made with a fast Fourier transform window length of 256 s and 50% overlap. (g) Dots indicate the dominating back azimuth at JO array in each 18 s long time window (f-k analysis section 3.1.1) colored according to slowness. Green horizontal lines respectively mark the back azimuth of signals from the eastern Skaftá cauldron, D15 and the point where the Skaftá river emerges from under the glacier (see Fig. 1). The black curve shows changes in back azimuth at JO corresponding to a point migrating along the flood path (Fig. 1) with a constant velocity 2 km/h passing D15 at 17:30 on 30 September. (h) Same as subfigure g but for IE array.

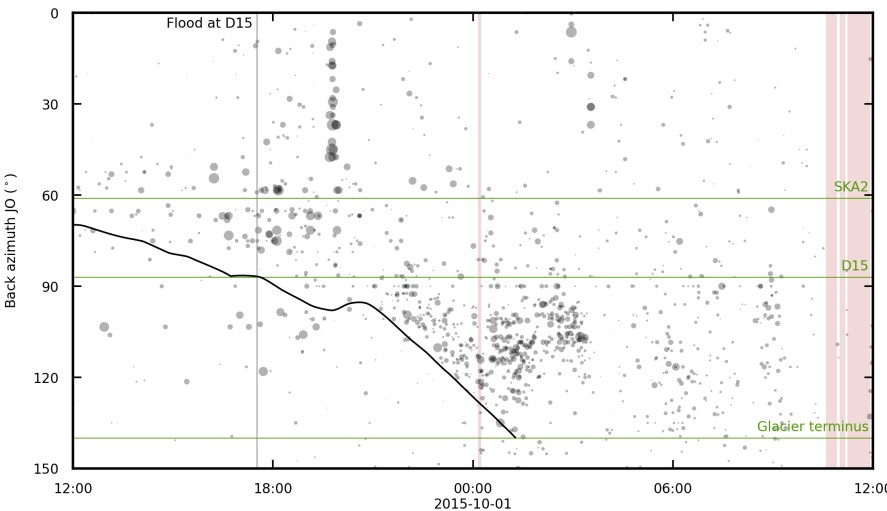

**Figure 3.** Transients detected by beam stacking on data filtered from 5 to 20 Hz (section 3.1.1). The size of the points indicates the semblance where all values are above 0.35 and slownesses are between 0.1 and 0.3 s/km (P waves only). Datagaps are highlighted with lightred background. The vertical gray line indicates the flood arrival at D15 (17:31 on 30 September). Black and green lines as in Fig. 2g.

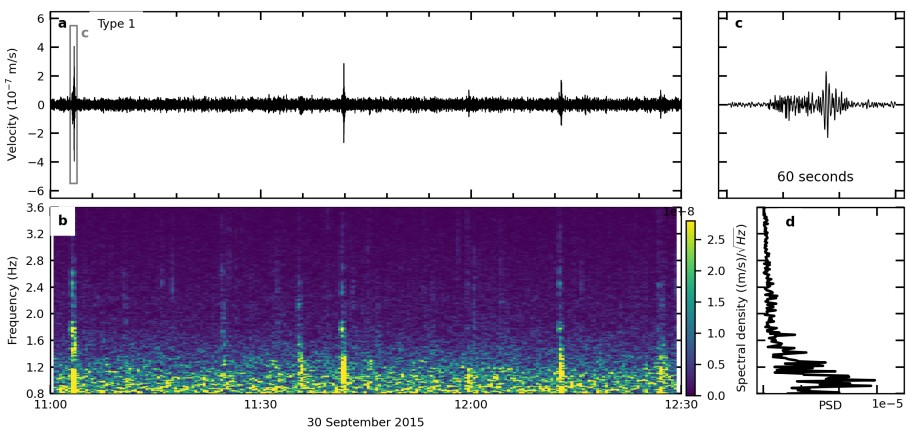

**Figure 4.** Icequakes followed the start of the flood. (a) Seismogram between 11:00 and 12:30 on 30 September 2015 filtered between 0.8 and 3.6 Hz, (b) amplitude spectrogram made with a fast Fourier transform window length of 64 s and 70% overlap. (c) 1 minute long time window showing a discrete event filtered between 1 and 8 Hz and (d) spectrum of subfigure b.

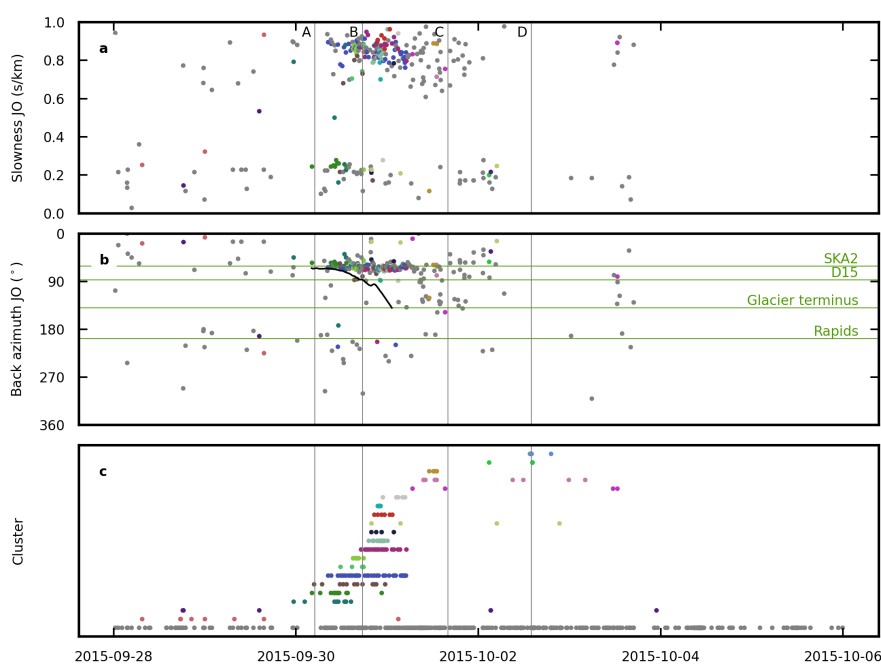

**Figure 5.** Icequakes detected with STA/LTA and template matching (section 3.1.3) and colored according to cluster membership (section 3.1.4). Unclustered events are shown in gray. Vertical gray lines indicate (A) the start of the flood (5:00 on 30 September), (B) the flood arrival at D15 (17:31 on 30 September), (C) the first levelling of the ice-shelf at SKA2 (16:00 on 1 October) and (D) the start of the final ice-shelf subsidence (14:00 on 2 October). (a) Slowness and (b) back azimuth at JO. The horizontal green lines mark from top to bottom the direction towards the cauldron, D15, the ice terminus and the rapids. Black line as in Fig. 2g. (c) Clusters sorted according to time of first occurrence.

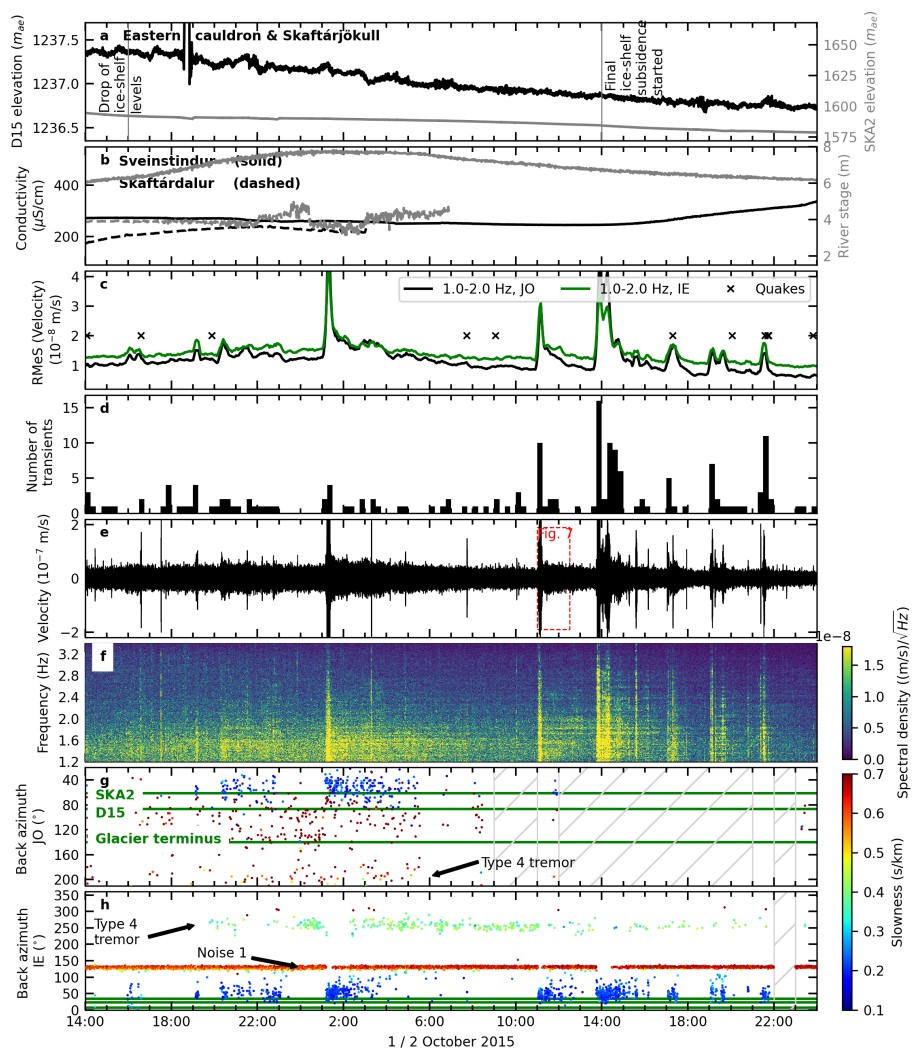

**Figure 6.** Type 2 tremor bursts and Type 3 harmonic tremor tails occurred after most of the water had drained from the subglacial lake. Tremor from the cauldron area from 14:00 on 1 October to 22:00 on 2 October 2015. Subfigures as in Fig. 2.

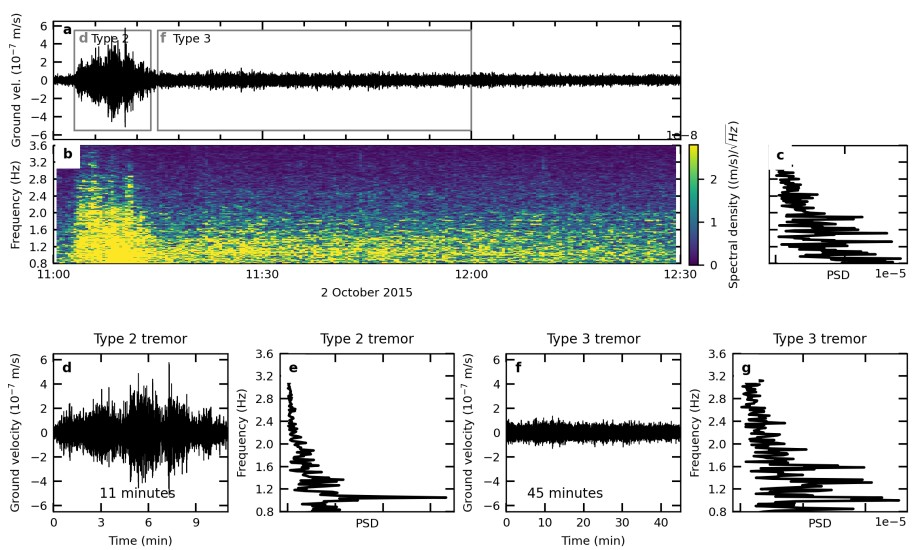

**Figure 7.** Zoom into one Type 2 tremor burst and its Type 3 harmonic tremor tail from 11:00 to 12:30 on 2 October 2015. Subfigures a and b as in Fig. 4 but in (b) with a fast Fourier transform window length of 84 s. (c) spectrum of subfigure b. (d) 11-minutes-long time window showing a zoom of the tremor burst and (e) its spectrum. (f) 45-minutes-long time window showing the harmonic tremor tail and (g) its spectrum.

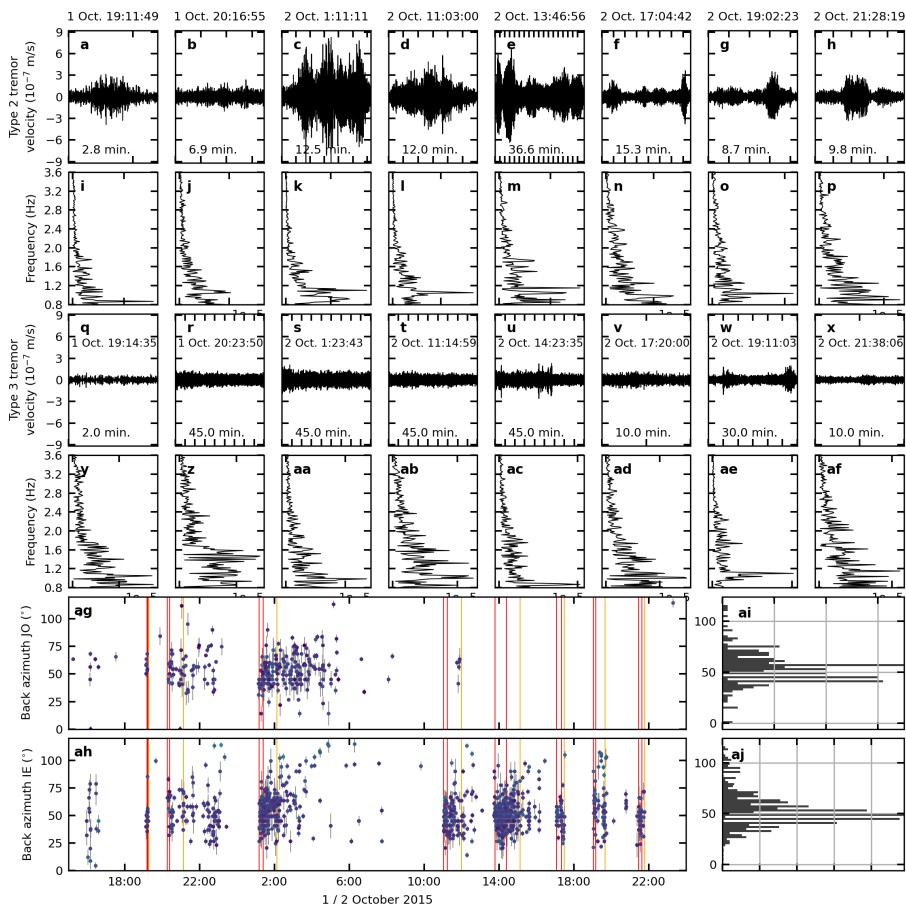

**Figure 8.** Temporal evolution of Type 2 and 3 tremor. (a-h) Vertical component seismograms of JOK filtered between 0.8 and 3.6 Hz for eight Type 2 tremors. Start time and duration of the time windows are given above and below the seismogram, respectively. (i-p) Spectra of seismograms in subfigures (a-h). (q-x) Same as subfigures (a-h) for 8 Type 3 tremors and (y to af) Spectra of seismograms in subfigures (q to x). (ag-ah) Back azimuths associated with the Type 2 and Type 3 tremor bursts at (ag) JO and (ah) IE array. Red and orange vertical lines indicate the time windows of Type 2 and Type 3 tremor shown in subfigures a to af. (ai-aj) Histograms illustrate the dominant back azimuths at (ai) JO and (aj) IE array that indicate a source in the cauldron area.

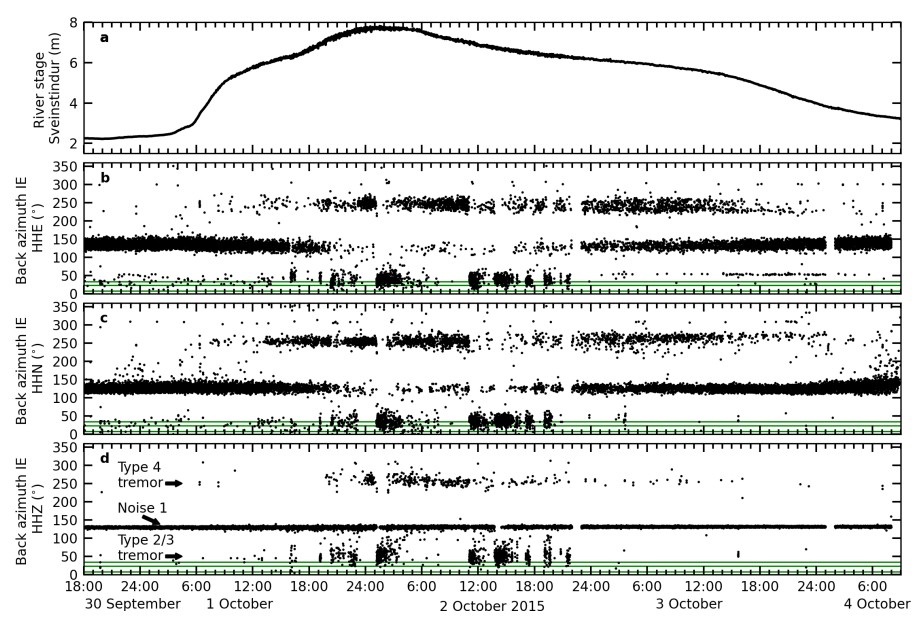

**Figure 9.** Comparison between river stage and Type 4 tremor recorded at IE array. (a) River stage in Sveinstindur. (b-d) Back azimuth of IE array derived using the (b) HHE, (c) HHN and (d) HHZ component of the seismometers. Horizontal green lines as in Fig. 2h.

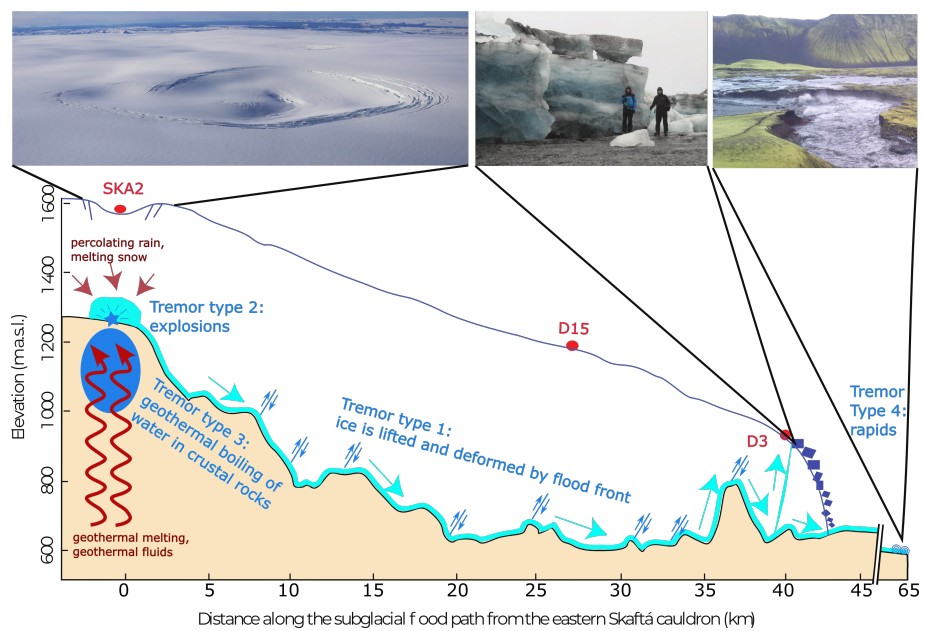

**Figure 10.** Schematic diagram along the subglacial flood path from the eastern Skaftá cauldron based on figure 3.6 in Einarsson (2009), figure 9 in Magnússon et al. (2021) and figure 2 in Jóhannesson et al. (2007). Conceptual model of the tremor generation supported by the high-frequency transients and clustered icequakes. The photos show (left) an aerial view from the east of the eastern Skaftá cauldron after the September/October 2015 jökulhlaup. The western cauldron can be seen in the distance to the right of the center of the image. Semi-circular crevasses mark the boundary of the area that subsided. Thrust ridges due to inward ice flow have formed near the center of the depression during the 10 days that had elapsed since the flood to the time when the photo was taken. Photo: Oddur Sigurðsson, 10 October 2015. (middle) Large ice blocks broken from the glacier surface by the initial outburst flood through the glacier at 3 km distance from the ice margin. Photo: Tómas Jóhannesson, 1 October 2015. (right) Rapids near Sveinstindur. Photo: Bergur Einarson, 2 October 2015.

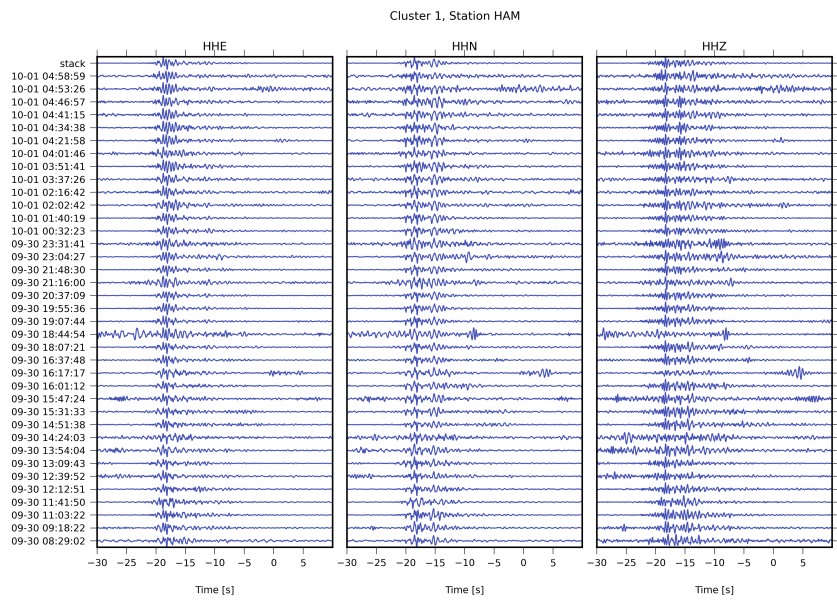

**Figure A1.** Example comparison of waveforms of events in cluster 1 as observed at station HAM.

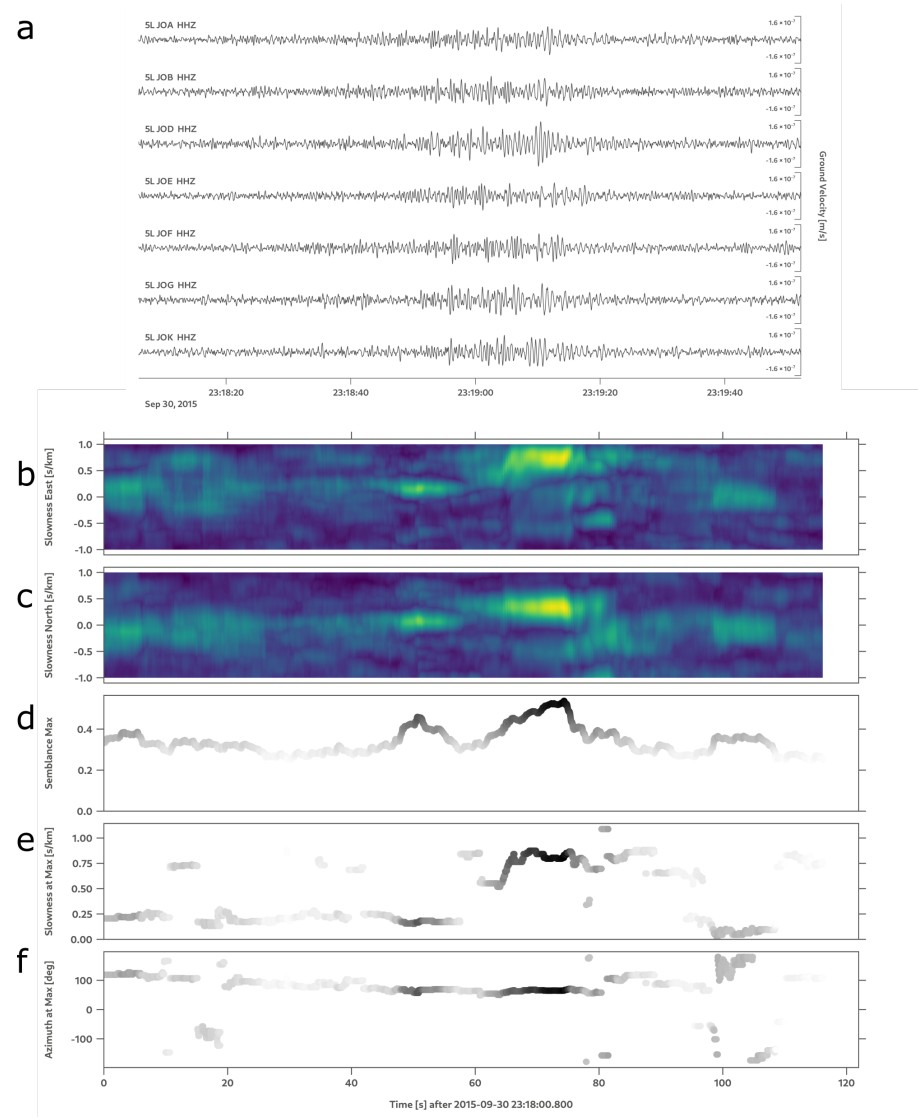

**Figure A2.** Example for a clustered (cauldron) event filtered 1.5 to 5 Hz. (a) Seismic waveforms of all stations in the JO array. (b-f) Beam-stacking grid search results as a function of time, top to bottom: (b) and (c) component-wise projection of semblance maximum, where lighter colors indicate higher semblance. (d) Maximum semblance over all slowness values. (e) Slowness and (f) back-azimuth values at semblance maximum value at each time step, where dark colors indicate higher corresponding semblance values.

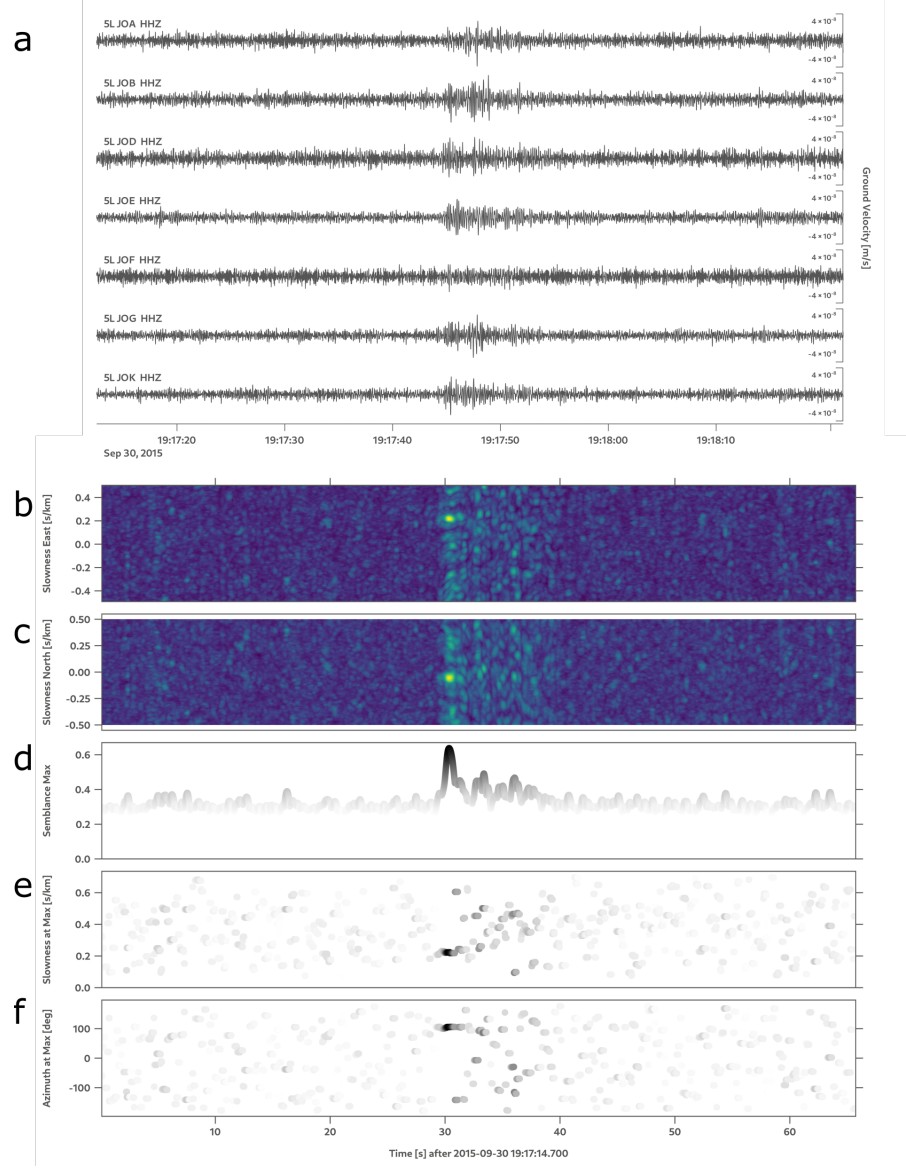

**Figure A3.** Example for a high frequency transient filtered 5 to 20 Hz. Same subfigures as in Suppl. Fig. A2.