# Peer review of "Subaerial and subglacial seismic characteristics of the largest measured jökulhlaup from the Eastern Skaftá cauldron, Iceland"

_EGUsphere, 2022_

## Author Response (AR1)

*Reply to all reviewers:*

*We thank the reviewers for their considered and insightful review. We agree with the reviewers that the processes are interesting and highly relevant for the glaciological and volcanological community when it comes to further our understanding of hazardous flooding events but also the heat exchange between rock and ice.*

*Based on the feedback, we recognize that we did not draw sufficient attention to the new insights that this manuscript is focusing on in comparison to Eibl et al. 2020. To improve the manuscript and address the reviewer's concerns, we have improved clarity throughout in the old text and figures, to draw attention to the new findings and remove duplication.*

*We have also conducted new processing of the seismic data from the SIL network in combination with the seismic arrays at JO and IE. We have now processed (i) all three components of the arrays. (ii) We performed template matching and clustering of additional icequakes that indicate the collapse of the ice cap above the lake. (iii) We detected high frequency transient events in the array data that support our interpretation that Type 1 tremor is composed of repeated events. (iv) We investigated the 'local noise' in more detail and located the sources at rapids and a nearby waterfall.*

*In comparison to Eibl et al. 2020, we here show that we can differentiate 4 instead of 2 different tremor types. Our detailed seismological analysis yields 3 different subglacial tremor types (Type I to III), rapid-induced tremor (Type 4) from subaerial river flow, icequakes originating in the cauldron area, and high-frequency transient events propagating alongside Type I tremor with the flood front. This study hence provides much more detail of the detected seismological signals and their generation associated with the subglacial but also subaerial propagation of the flood.*

*As recognized by the reviewers we would like to help to classify tremors in less studied areas with this detailed and extensive study of a robust dataset. Further details of our modifications can be found in bullet points below:*

**Reviewer 1:**

**Review to "Seismic Characteristics of the Largest Measured Subglacial Flood from the Eastern Skaftá cauldron, Iceland" by Eibl and others, 2022**

The authors of the manuscript present seismological observations from a subglacial flood that originated in 2015 from the Eastern Skaftá cauldron at the Vatnajökull ice cap in Iceland. Over the course of a few days, they detect various quakes and two types of tremors which they exploit to study the temporal evolution of the flood by means of quake locations, beamforming analysis, spectrograms and tremor amplitude. Guided by complementary measurements of the ice motion and hydrological parameters, Eibl et al. find that the quakes are related to the subsidence of the cauldron and the two tremor types to the subglacial hydrology: the longer lasting type 1 tremor is attributed to the flood wave propagation while the shorter but more impulsive type 2 tremor is attributed to geothermal processes including boiling in response to the flood. The study focuses on type 1 tremor and the authors suggest that this signal is caused by multiple brittle ice cracking as the flood wave propagates between ice and bedrock, lifting

the glacier by up to 1 m. This tremor type can be used to track the propagation of the flood and its speed via beamforming analysis (Eibl et al. 2020).

The investigated processes are interesting and highly relevant for the glaciological(/volcanological) community. Furthermore, the results appear robust and the conclusions convincing. However, in my opinion, the submitted manuscript does not provide significant new data and/or processing and thus not significant novel insights into the flood. Most of the eight figures show material, that is already presented in Eibl et al. (2020) (e.g. most of Figs. 1, 3, 4, 6) and the conclusion drawn are the same for both manuscripts. Doubtlessly, the present manuscript contains a more in-depth discussion on the involved processes compared to Eibl et al. (2020), but still concludes that type 1 tremor is caused by the propagating flood wave lifting the ice and type 2 tremor by hydrothermal explosions and subsequent geothermal boiling. Only the quake and water-chemical data are newly introduced but these solely play a side role and do not provide significant novel insights. Overall, the manuscript appears more like a supplementary material to Eibl et al. (2020). For this reason, I unfortunately cannot recommend to consider the article for publication in Earth Surface Dynamics.

**Reference**

Eibl, E. P. S., Bean, C. J., Einarsson, B., Pàlsson, F., & Vogfjörd, K. S. Seismic ground vibrations give advanced early-warning of subglacial floods. Nature Communications **11**, 2504 (2020). https://doi.org/10.1038/s41467-020-15744-5

*Reply:*

- *We performed a new array processing at higher frequencies and detected transient events that follow the flood front. This supports our conclusion that tremor Type 1 is composed of icequakes.*
- *We ran a STA/LTA detector and correlated the waveforms to detect more events. Our final catalog was clustered based on the station HAM, which was the closest 3-component seismometer to the sources based on our back azimuth estimate. Based on station HAM, the waveforms were clustered into 20 different families. These all originate in the cauldron area indicating a gradual collapse of the ice-shelf on different faults.*
- *We analyzed, located, and discussed the other detected seismic sources which are caused by the flood once it reaches the subaerial river (previously just referred to as "local noise" in Eibl et al 2020)*
- *We realized that we had not made it clear enough that we are discussing three different tremor types here (while Eibl et al. 2020 only discuss two)*
- *We modified all figures to highlight the above-mentioned new points (and deleted former Fig. 2 as this was merely doubling most information).*
- *We clarified that the role of the geochemistry is crucial when it comes to discussing the source of Type 2 and 3 tremor in the context of volcanic eruptions or hydrothermal explosions.*
- *We modified the discussion to add our new insights about the drainage processes.*
- *Based on our additional analysis we modified our abstract and conclusion.*

**Reviewer 2:**

This preprint describes the largest measured subglacial flood from the Eastern Skafta cauldron in Iceland in 2015. The Authors aim to improve the current understanding of processes behind seismic signal generation during subglacial floods. Thanks to the analysis of seismic, GPS, and hydrological observations, the Authors propose two source mechanisms from tremor signal generation: geothermal boiling of water in crustal rocks and repeating icequakes caused be glacier lift.

Yet, most of these observations and the same event have been already published in the paper by Eibl et al., 2020. Moreover, the Authors used the same methods to analyze seismic data. I believe that for this paper to be published, more new information or novel processing approaches should be explored. Some of the claims seem speculative now; for example, the authors propose that tremor 1 is associated with repeating icequakes. This can be very easily verified with clustering methods (e.g., RedPy, Hotovec-Ellis et al., 2019) or template matching (Beaucé et al., 2018). For now, I do not see much value added and novelty compared to Eibl et al. 2020 paper, which, unfortunately, does not allow me to accept this preprint.

References:

Beaucé, E., Frank, W. B., and Romanenko, A.: Fast Matched Filter (FMF): An Efficient Seismic Matched-Filter Search for Both CPU and GPU Architectures, Seismological Research Letters, 89, 165–172, https://doi.org/10.1785/0220170181, 2018

Eibl, E.P.S., Bean, C.J., Einarsson, B. et al. Seismic ground vibrations give advanced early-warning of subglacial floods. Nat Commun **11**, 2504 (2020). https://doi.org/10.1038/s41467-020-15744-5

Hotovec-Ellis, A. and Jeffries, C.: Near Real-time Detection, Clustering, and Analysis of Repeating Earthquakes: Application to Mount St. Helens and Redoubt Volcanoes, in: Presented at Seismological Society of America Annual Meeting, 2016

***Reply:**_*

- *We performed a new array processing at higher frequencies and detected transient events that follow the flood front. This supports our conclusion that tremor Type 1 is composed of icequakes.*
- *We tested RedPy, but the clustering based on a predefined transient list did not yield any families. We hence ran a STA/LTA detector and correlated the waveforms ourselves to detect more events. Our final catalog was clustered based on the 3-component station HAM, which was closest to the sources based on our back azimuth estimate. Based on station HAM, the waveforms were clustered into 20 different families. These all originate in the cauldron area indicating a gradual collapse of the ice-shelf on different faults.*
- *We analyzed, located, and discussed the other detected seismic sources which are caused by the flood once it reaches the subaerial river (previously just referred to as "local noise" in Eibl et al 2020)*
- *We realized that we had not made it clear enough that we are discussing three different tremor types here (while Eibl et al. 2020 only discuss two)*
- *We modified all figures to highlight the above-mentioned new points (and deleted former Fig 2 as this was merely doubling most information).*
- *We modified the text severely to reflect the additional processing we have done.*

**Reviewer 3:**

The manuscript "Seismic Characteristics of the Largest Measured Subglacial Flood from the Eastern Skaftá cauldron, Iceland" presents seismic, GPS and hydrological data, collected during a subglacial flood event in Iceland. The authors found two different types of glacial tremors. Both are related to the flood event. Type 1 tremor originates due to the glacier uplift, leading to icequakes. In the manuscript, the authors show the propagation of the tremor towards the glacier terminus by calculating the back azimuth and comparing these results with GPS and hydrological measurements. Additionally, to the tremor caused by icequakes (type 1), a second tremor type caused by boiling water, called type 2, is presented. After emptying the subglacial lake, the pressure at the glacier bed decreases and allows the water in the volcanic hydrothermal system to boil. Seismometers can see the exploding water bubbles, which show up as type 2 tremor. The authors also present the typical frequency band of both tremor types.

The study of interactions between solid, fluid and gaseous water is unique. In my opinion, the topic of this manuscript is important to understand the hazardous flooding events but also the heat exchange between rock and ice better. Nevertheless, to me the manuscript does not include enough new findings, compared to Eibl et al., 2020, to be presented as a self-standing paper. The manuscript seems to me more like supplementary information to Eibl et al., 2020 because most of the figures and the results are already presented in Eibl et al., 2020. Based on my findings, I do not recommend this manuscript for publication in the current state.

Adding new methods to the same dataset would help to make this manuscript a self-standing paper. For example, Eibl et al., 2020 proposed an early-warning approach using tremor type 1, which could be tested in this manuscript. Increasing the warning time and reaching as few false alarms as possible would be a beneficial application of the findings shown. Another further application would be to train machine learning algorithms to cluster the tremor types automatically by analyzing different tremor features in the time and frequency domain. The seismic dataset seems robust and can help to classify tremors in less studied areas. Also trying to reproduce the measured seismic data with a physical model would improve the manuscript and make it a self-standing paper. A physical model would help to understand the complicated processes at the glacier ice, glacier bed, and englacial lake interface, including the volcanic heat fluxes. However, based on the above-mentioned reasons I can not recommend accepting this manuscript for publication in Earth Surface Dynamics.

Reference:

Eibl, E. P. S., Bean, C. J., Einarsson, B., Pàlsson, F., & Vogfjörd, K. S. Seismic ground vibrations give advanced early-warning of subglacial floods. Nature Communications 11, 2504 (2020). https://doi.org/10.1038/s41467-020-15744-5

***Reply:***

- *We think that testing the early-warning approach would bring this manuscript closer to Eibl et al. 2020 and hence did not implement this.*
- *We also think that a sample size of 4 is too small to implement a systematic early warning approach or machine learning.*
- *However, we performed a new array processing at higher frequencies and detected transient events that follow the flood front. This supports our conclusion that tremor Type 1 is composed of icequakes.*
- *We ran a STA/LTA detector and correlated the waveforms to detect more events. Our final catalog was clustered based on the station HAM, which was the closest 3-component seismometer to the sources based on our back azimuth estimate. Based on station HAM, the waveforms were clustered into 20 different families. These all originate in the cauldron area indicating a gradual collapse of the ice-shelf on different faults.*

- *We analyzed, located, and discussed the other detected seismic sources which are caused by the flood once it reaches the subaerial river (previously just referred to as "local noise" in Eibl et al 2020)*
- *We realized that we had not made it clear enough that we are discussing three different tremor types here (while Eibl et al. 2020 only discuss two)*
- *We modified all figures to highlight the above-mentioned new points (and deleted former Fig 2 as this was merely doubling most information).*
- *We modified the text throughout to reflect these additions.*

---

## Referee Report (RR1)

Review of Eibl et al. – Esurf

In fairness to the authors and in keeping with prior discourse on this manuscript, I have tried to focus my review on how well the authors have addressed the issues that all three previous reviewers identified. In broad strokes, I would agree with their assessment. The dataset is interesting and offers some intriguing possibilities toward understanding outburst flood processes, hydrothermal/ice interactions, and hazard early warning. I am not a seismologist and am approaching this from a geomorphologist's perspective so I can't offer too much on methodology, and the previous reviewers seem to agree that it's all good.

I see some substantive improvement particularly in the discussion of tremor sources and origins. I don't know that I would say that splitting the type-2 tremors into 2 and 3 referring to the tremor generated by hydrothermal explosions and subsequent boiling is new, as the type-2 tremor is describes in Eibl et al. (2020) as likely being driven by both processes already, but in general, I think the additions make the manuscript a lot more distinctive from the 2020 paper. I see a few issues that I think the authors could address without much new analysis.

- First, I think there needs to be a little more methodological explanation. Coming from the outburst flood crowd rather than the environmental seismology crowd, I had to look up a lot of terminology and there are abbreviations (STA/LTA) that go undefined. For an audience of geomorphologists, I'd like to see a little more explanation of the methods and interpretation of the waveforms, etc.

- I'm not sure I fully understand what new insights the clustering analysis brought to the table. I see the authors have attributed to the clustering of events into collapse of the ice shelf, but it's unclear to me precisely why.

- (382) are the faults referred to here faults within the ice?

- There's no figure 1

- (Fig. 3) "beamforming" another example of a term that might be well known to folks working in seismology and signal processing but I haven't come across as a geomorphologist. For ESurf I think some of these terms just need clarification.

- (Fig 7c) lacks a scale

---

## Author Response (AR2)

*We thank the reviewers for their constructive feedback and modified the figures and text to address the comments. Further details can be found below.*

**Reviewer 1**

In the revised version of the manuscript, the Authors provide an additional analysis of the signals seismic signal generation during the largest measured subglacial flood from the Eastern Skafta cauldron in Iceland in 2015.

Now the manuscript provides some additional information compared to Eibl et al., 2020. However, the description of different processes associated with subglacial floods and the corresponding seismic signals is not clear. Also, the description of methods could be more precise, especially since the method section is mixed with the results. In general, the manuscript is interesting but not easy to follow, and I suggest that the Authors spend some time clarifying and simplifying specific parts of the manuscript.

I have a few major comments. First, it needs to be clarified how the Authors divide the seismic signals into different types of tremors. The Authors use different seismic metrics (e.g., back azimuth, slowness, frequency content) combined with ancillary measurements (e.g., GPS, hydrological) to classify the seismic signals into tremors 1, 2, 3, and 4. However, this classification seems somehow arbitrary. I believe it would be easier for the Reader to follow the manuscript if the Authors first introduced different metrics they use to describe the seismic signals and constrain their sources; based on those metrics and ancillary data, divide seismic signals into different types of "tremors," and then finally interpret the tremors in terms of source processes generating them.

***Reply***: *We added a table to clarify the tremor properties and terms used. We also added a sentence to clarify the metrics used to separate the tremor into different types.*

Then, the methods, results, and interpretation are all mixed. Certain parts of the manuscript could be shortened (e.g., 5.2.1), and a certain part gives repeating information (see below for details). Moreover, the Authors used multiple terms to describe the same methods (e.g., array processing, beamforming, FK…) and the same processes (e.g., quakes, icequakes, and transient events)-maybe their signals have different source processes, but for now, it is very hard to keep track of which signals are generated by what.

***Reply***: *We added a table to clarify the tremor properties and terms used. We double checked our usage of "transients" and now follow strictly our definition. See below for further details.*

Then, some terms do not seem fully accurate. For example, the Authors call tremor 3 a harmonic tremor. The spectrogram in Figure 7b does not show any spectral lines that I would expect to see in a harmonic tremor. Also, the amplitude peak in PSD in panel g for the dominant frequency does not seem very pronounced compared to panel e (non-harmonic termor, I assume?).

***Reply***: *We modified the way we calculate the spectra. Now we calculate spectra for 1 min long seismic data. For windows longer than 1 minute, we stack the resulting spectra and only plot the final spectrum. This enhances coherent frequencies and here the harmonic character.*

Also, what is the cross-correlation coefficient between "transient events"? 0.2? If yes, you cannot really say these are repeating events or event multiplets… See Uchida and Bürgmann (2019).

***Reply***: *Higher cross correlation values are not reached because both template and matched event are noisy, the noise level at some stations are higher and the length of the template is longer than the event's signal. The average background level of the cross-correlation of the event templates with the continuous waveforms is on the order of 0.015. Against this, a value of 0.2 is highly significant and only reached when signals of both events arrive at all contributing stations and components without any differential time delays and also only when the coda of both events is qualitatively very similar (Fig. S1). We cannot say whether the events are truly repeating or if they occur in close neighborhood of each other. The shortest contributing wavelengths are on the order of about 130 m. We edited the text.*

See below for more comments:

Line 40: When studying… - Why do those different sources need to be characterized? Could you add some more information on the impact of this characterization?

***Reply***: *Done*

Line 65: Could you give the exact days?

***Reply***: *Done*

Line 114: Is it possible that those 45 events are the events that you also detect through STA/LTA and template matching?

*Reply: 39 of those 45 events are also detected through our STA/LTA and template matching. 22 are sorted into clusters and are hence likely icequakes, while the other ones are possibly earthquakes.*

Line 128: If you use the horizontal components, the FK should be performed on rotated components, R and T; otherwise, the slowness measurements are affected. Also, do you only show the results for the vertical component in the manuscript, or did I miss something?

*Reply: Yes, we only show the vertical component in the manuscript. Since we face a migrating source, we did not rotate to r and t. We have clarified this in the text.*

Line 131: At less… -this should be in results

*Reply: We deleted this sentence which is very general and does not provide a lot of detail: 'At less than 50 km source--receiver distance, migrating subglacial flood-related tremor sources can be tracked with high temporal and spatial resolution, and changes in tremor source depth or wavetype can be detected.'*

Line 133: Define semblance.

*Reply: Semblance is defined as the ratio of the coherent energy to the total energy in the waveform stack within the time window of analysis (e.g. Kennett 2000).*

Line 139: This belongs in Results.

*Reply: We disagree. We describe here in general how the back azimuth results might look like and what that might mean without discussing our results yet.*

Line 147: That would fit better in Results again.

*Reply: Moved to results.*

Line 152: FK is also an array processing method… In fact, what you describe here is also referred in the literature as FK (the search over sx and sx, if I understand correctly). Could you show some results of that in an appendix?

*Reply: For the higher frequency array processing we used beam stacking in the time domain as an alternative to FK analysis. Though very similar to FK, it is more efficient for very short time windows. We clarified this in the text and through new headings. We added an example of such a detection to the appendix.*

Line 170: on the lower half of the flood propagation path-not clear

*Reply: deleted*

Line 180: The threshold of 0.2 seems too low to claim a similarity of waveforms.

*Reply: See reply above and modification in the manuscript.*

Line 189: the closest to the flood path?

*Reply: yes. Modified.*

Line 199: We selected…-this sentence is very confusing. You did not mention beam power before.

*Reply: The result of the FK analysis are beam power, semblance, slowness, and azimuth for processing time windows of 18 seconds, as described in section 3.1.1. When multiple such time windows intersected with the time interval of the signal detected through STA/LTA and template matching, we chose the azimuth and slowness values from the time window with the highest product of beam power and slowness. To avoid the confusion we decided to remove this detail from the text.*

Line 214: Result and Interpretation-isn't Interpretation Discussion?

*Reply: We provide a detailed literature discussion on the topic in the 'discussion' section and keep the literature in the results and interpretation section to a minimum. We hence kept the labelling as 'results and interpretation' and "discussion".*

Line 232: a significant amount-could you specify?

*Reply: We clarified that we detected 40-100 transients per hour at that time (semblance values > 0.35).*

Line 239: conductivity measurements were not introduced yet?

*Reply: The conductivity measurements were mentioned in section 3.2. of the methods.*

Line 246: Quakes or icequakes?

*Reply*: *These might be icequakes or earthquakes. We clarified the term in the new table 1.*

Line 291: You already said it.

*Reply*: *We removed the first part of the sentence.*

Line 295: Again, are those events really "repeating" and highly similar? What is the cross-correlation ratio between these events?

*Reply*: *See reply above and new figure S1.*

Line 299: similar characteristics-such as?

*Reply*: *We clarified in the text: "showed qualitatively similar waveforms in frequency content, duration and waveform pattern when inspected visually"*

Line 305: in the range of 0.8 to 0.8 s/km

*Reply*: *Sorry for this typo. Corrected to 0.7 to 0.9.*

Lines 387: 392-this has already been said in Introduction.

*Reply*: *We deleted this paragraph.*

Figure 1: Gray vertical line at 20h; what does it denote?

*Reply*: *There is no grey line in Fig. 1. We removed the grey line in Fig. 2.*

Figure 1: Can some of those quakes also be classified as transients? Could you show some waveforms?

*Reply*: *We added one example for a clustered (cauldron) event filtered 1.5 to 5 Hz and a high frequency transient, 5 to 20 Hz in the supplementary material.*

The black curve shows forward-calculated(?) changes…

*Reply*: *The black curve shows changes in back azimuth at JO corresponding to a point migrating along the flood path (Fig. 1) with a constant velocity 2 km/h passing D15 at 17:30 on 30 September.*

Figure 3: You do not mention beamforming in the text.

*Reply*: *We replaced it with 'beam stacking' and use the same term, as in the methods section. We annotated the grey and green lines.*

Figure 5: Could you annotate the vertical gray lines? Any comments on the change between slowness values of different clusters? The one appearing earlier have low slowness values ~0.2 s/km; could they be coming from depth?

*Reply*: *The events show body waves at ~0.2 s/km followed by surface waves at ~0.8 s/km. The FK analysis with 18 second time windows cannot separate the two peaks and picks the higher one. Which one is selected depends on the filter settings and the frequency content of the background noise. For example, if we run the FK analysis including higher frequencies more of the events move to 0.2 s/km slowness. We added waveforms of an event as supplementary material.*

Also, panel c shows a progressive activation of different clusters. That seems interesting. Could you comment on this?

*Reply*: *We agree that this is interesting. This might reflect the slip along different fault planes in the ice, when the ice-shelf gradually subsided. In our opinion this supports an icequake interpretation rather than an earthquake interpretation. We clarified this in the text.*

Figure 7: As I already said, is this really a harmonic tremor? Could you try to improve the quality of the spectrogram in b to check if any spectral lines are visible?

*Reply*: *We improved the quality of the spectra (see above). The peaks are now visible.*

Panels c, d, e, f, and g are missing labels and units (c, e, and g).

*Reply*: *We added the missing labels and units.*

Figure 8: Again, the amplitude peaks at dominant frequencies are not visible for the "harmonic" tremor.

*Reply*: *We improved the quality of the spectra (see above). The peaks are now visible.*

Uchida, N., & Bürgmann, R. (2019). Repeating earthquakes. Annual Review of Earth and Planetary Sciences, 47 (1), 305-332. Retrieved from https://doi.org/10.1146/annurev-earth-053018-060119 doi: 10.1146/annurev-earth-053018-060119

**Reviewer 2**

Review of Eibl et al. – Esurf

In fairness to the authors and in keeping with prior discourse on this manuscript, I have tried to focus my review on how well the authors have addressed the issues that all three previous reviewers identified. In broad strokes, I would agree with their assessment. The dataset is interesting and offers some intriguing possibilities toward understanding outburst flood processes, hydrothermal/ice interactions, and hazard early warning. I am not a seismologist and am approaching this from a geomorphologist's perspective so I can't offer too much on methodology, and the previous reviewers seem to agree that it's all good.

I see some substantive improvement particularly in the discussion of tremor sources and origins. I don't know that I would say that splitting the type-2 tremors into 2 and 3 referring to the tremor generated by hydrothermal explosions and subsequent boiling is new, as the type-2 tremor is describes in Eibl et al. (2020) as likely being driven by both processes already, but in general, I think the additions make the manuscript a lot more distinctive from the 2020 paper. I see a few issues that I think the authors could address without much new analysis.

- First, I think there needs to be a little more methodological explanation. Coming from the outburst flood crowd rather than the environmental seismology crowd, I had to look up a lot of terminology and there are abbreviations (STA/LTA) that go undefined. For an audience of geomorphologists, I'd like to see a little more explanation of the methods and interpretation of the waveforms, etc.

*Reply: We added a definition for STA/ LTA and semblance in the text. We also added a table to clarify the terms we used and made sure to use them consistently throughout the text. We also explain terms when first mentioning them e. g. seismic array. Finally, we added a sentence in the Introduction Chapter, specifically referring to the paper by Podolskiy and Walter (2016) which provides a really good, and comprehensive overview of seismic sources and seismic methods commonly used in cryoseismology.*

- I'm not sure I fully understand what new insights the clustering analysis brought to the table. I see the authors have attributed to the clustering of events into collapse of the ice shelf, but it's unclear to me precisely why.

*Reply: We have rephrased the text to clarify.*

- (382) are the faults referred to here faults within the ice?

*Reply: Yes. We modified the text to clarify.*

- There's no figure 1

*Reply: Figure 1 is in the text, while figure 2 ff is below the text.*

- (Fig. 3) "beamforming" another example of a term that might be well known to folks working in seismology and signal processing but I haven't come across as a geomorphologist. For ESurf I think some of these terms just need clarification.

*Reply: We replace beamforming with 'beam stacking'. This method is described in more detail in the methods.*

- (Fig 7c) lacks a scale

*Reply: We added the missing units and labels to the subfigures.*